# Automated bone mineral density prediction and fracture risk assessment using plain radiographs via deep learning

Chen-I Hsieh [1], Kang Zheng [2], Chihung Lin[3], Ling Mei[4], Le Lu [2], Weijian Li[2], Fang-Ping Chen [5,6], Yirui Wang [2], Xiaoyun Zhou [2], Fakai Wang [2], Guotong Xie [7], Jing Xiao [7], Shun Miao [2✉] & Chang-Fu Kuo [1,2,5✉]

Dual-energy X-ray absorptiometry (DXA) is underutilized to measure bone mineral density (BMD) and evaluate fracture risk. We present an automated tool to identify fractures, predict BMD, and evaluate fracture risk using plain radiographs. The tool performance is evaluated on 5164 and 18175 patients with pelvis/lumbar spine radiographs and Hologic DXA. The model is well calibrated with minimal bias in the hip (slope = 0.982, calibration-in-the-large = −0.003) and the lumbar spine BMD (slope = 0.978, calibration-in-the-large = 0.003). The area under the precision-recall curve and accuracy are 0.89 and 91.7% for hip osteoporosis, 0.89 and 86.2% for spine osteoporosis, 0.83 and 95.0% for high 10-year major fracture risk, and 0.96 and 90.0% for high hip fracture risk. The tool classifies 5206 (84.8%) patients with 95% positive or negative predictive value for osteoporosis, compared to 3008 DXA conducted at the same study period. This automated tool may help identify high-risk patients for osteoporosis.

---

[1] Division of Rheumatology, Allergy and Immunology, Chang Gung Memorial Hospital, Taoyuan, Taiwan. [2] PAII Inc., Bethesda, MD, USA. [3] Center for Artificial Intelligence in Medicine, Chang Gung Memorial Hospital, Taoyuan, Taiwan. [4] Wuhan Hospital of Traditional Chinese Medicine, Wuhan, China. [5] Department of Medicine, College of Medicine, Chang Gung University, Kwei-Shan, Taoyuan, Taiwan. [6] Department of Obstetrics and Gynecology, Osteoporosis Prevention and Treatment Center, Keelung Chang Gung Memorial Hospital, Keelung, Taiwan. [7] Ping An Insurance (Group) Company of China, Ltd., Shenzhen, Guangdong, China. ✉email: shwinmiao@gmail.com; zandis@gmail.com

Osteoporosis is a common bone disease[1] that increases the global health burden[2]. All major types of osteoporosis-related fragility fractures are associated with chronic pain, disability, functional dependence[3], high morbidity[4], and a two-fold to three-fold increase in mortality[5], despite the availability of effective anti-osteoporotic drugs[6]. Dual-energy X-ray absorptiometry (DXA) is the preferred modality for the measurement of bone mineral density (BMD) in the human hip or lumbar spine, which is an essential component of the fracture risk assessment tool (FRAX) used to estimate the 10-year risk of hip or major osteoporotic fracture[7]. According to International Osteoporosis Foundation, DXA is falling short of the minimum service requirement for DXA of 11 units per million population[8] in most parts of Eastern Europe and Central Asia[9], the Middle East and Africa[10], Asia Pacific[11] and Latin America[12], and ten member states in European Union[13]. The US is well-resourced but both DXA and FRAX seem underutilized[14]. Among Medicare beneficiaries ≥65 years of age, only 30% of women and 4% of men were tested for BMD with DXA[15]. Among people with fragility fractures, only 10.2% of female[16] and 6% of male patients[17] have undergone BMD testing before the index event. Furthermore, DXA utilization seems to be decreasing in post-menopausal women in the US[18].

Opportunistic screening for osteoporosis using imaging modalities other than DXA is a potential strategy to stratify the unscreened population into distinct risk groups of osteoporosis and fragility fractures. This approach used radiographs 'already been taken' for other clinical indications to screen osteoporosis at no additional cost, time, or radiation exposure to patients. For example, several studies used computed tomography (CT)-based metrics to estimate BMD[19–21], classify osteoporosis[22], simulate DXA $T$-scores[23], and predict fracture outcomes[24]. However, the performance, radiation dose, and population coverage of CT-based screening strategies are barriers to their use in clinical settings. Unlike DXA and CT, plain radiography has greater availability, broader indications, lower radiation dose, and overall costs. In addition, the spatial resolution of radiographs is excellent, allowing the visualization of fine bone texture, which is correlated with bone density[25] and can distinguish patients with osteoporotic fractures from controls[25–27]. Therefore, an automated tool based on hip or spine radiographs for identifying hip fracture and vertebral compression fracture (VCF), predicting BMD, and evaluating fracture risk can help identify patients with greater fracture risk among individuals undergoing radiography of the hip or spine for other reasons.

Deep learning algorithms have achieved performance superior to traditional methods in visual recognition tasks[28], which is the foundation of clinical applications such as fracture detection[29], retinopathy grading[30], and lung nodule identification[31]. Therefore, this retrospective cohort study was performed to test the hypothesis that an automated deep neural network-based tool could effectively predict BMD and risk of fragility fractures using plain radiographs of the pelvis and lumbar spine. Here, we proposed and validated a fully automated deep learning-based tool to 1. extract the hip and spine region of interests (ROIs), 2. identify hip fracture, VCF, or morphological abnormalities, 3. check the radiograph quality to ensure that implants and foreign bodies were absent from the ROIs, 4. predict BMD and estimate the probability of a fracture within the next 10 years based on the FRAX (Fig. 1). We compared the predicted BMD with the BMD measured by central DXA. We also compared the risks of 10-year hip and major osteoporotic fractures (https://www.sheffield.ac.uk/FRAX/) using only clinical parameters (FRAX-NB, age, sex, weight, and height), clinical parameters and DXA-measured BMD (FRAX-MB) or predicted BMD (FRAX-PB). We also conducted a real-world test on consecutive patients to prove the clinical applicability of our tool, and its impact on osteoporosis screening strategy.

## Results

**Data source**. From 2006 to 2020, 30,958 and 86,977 patients aged 40–90 years with paired DXA-pelvis or paired DXA-lateral radiographs of the lumbar spine (18.6% and 18.2% of patients with hip or lumbar spine radiographs) were screened to identify hip and spine cohorts for analysis. Of these, 18,097 and 58,149 patients in the respective cohorts were excluded due to a DXA-radiograph interval >180 days, lack of detailed reports, inadequate image quality, positions, or analyzable ROIs. The final cohorts included 10,797 patients with Hologic DXA-hip radiograph pairs and 25,482 patients with Hologic DXA- spine radiograph pairs (Supplementary Fig. 1). No patient was included in more than one group. As Table 1 shows, the final study population included 5164 patients (3997 women [77.4%], mean age, 72.2 [standard deviation, SD, 11.2] years) in the hip testing set and 18,175 patients (14,469 women [79.6%], mean age, 67.1 [SD, 10.6] years) in the spine testing set. The median time between DXA and plain radiographs was 29 and 16 days, respectively. The DXA identified 1110 patients (21.5%) in the hip and 7860 patients (43.3%) in the spine cohort as osteoporotic.

**Performance for BMD prediction**. Table 2 summarizes the model performance to predict BMD. Pearson's correlation coefficients between DXA-measured and model-predicted BMD were 0.92 for the hip and 0.90 for the lumbar spine. The linear regression model showed excellent predictive performance of predicted BMD with regard to measured BMD (hip: $R^2 = 0.84$, root mean square error [RMSE] = 0.062; spine: $R^2 = 0.81$, RMSE = 0.081). The model was well calibrated with minimal bias in the hip (slope = 0.982, calibration-in-the-large = −0.003) and the lumbar spine BMD (slope = 0.978, calibration-in-the-large = 0.003) (Fig. 2). The model performance remained robust across various age and sex strata.

**Performance for osteoporosis and fracture risk prediction**. Table 3 illustrates the discriminatory performance of the model to classify hip or spine osteoporosis, and identify patients with greater 10-year risks of major osteoporotic fractures (≥20%) and hip fractures (≥3%). The algorithm provided a high degree of discrimination for osteoporosis (area under the precision-recall curve [AUPRC], 0.89 for both the hip and spine models). The overall accuracies were 91.7% for hip osteoporosis and 86.2% for lumbar spine osteoporosis. The median FRAX 10-year major fracture (8.84% vs. 8.76%, $p = 0.24$) and hip fracture risks (2.48% vs. 2.46%, $p = 0.06$) did not significantly differ, when scores were based on the predicted BMD (FRAX-PB) or measured BMD (FRAX-MB) plus clinical parameters (age, sex, height, and weight). The area under the precision-recall curve (AUPRC) values for major osteoporotic and fractures were 0.83 and 0.96 for (FRAX-PB), compared to 0.40 and 0.83 for the FRAX tool without BMD input (FRAX-NB) (Supplementary Figs. 2 and 3). Supplementary Table 1 shows robust model discriminatory performances across age and sex strata.

Next, we identified predicted BMD thresholds that correspond to a positive predictive value (PPV) of 95% to classify and a negative predicted value (NPV) of 95% to exclude osteoporosis (Table 4). Overall, 88.2% of the predicted values in the hip cohort and 70.4% in the spine cohort have an excellent PPV or NPV for osteoporosis. Among the hip cohort, FRAX-PB provides excellent discriminatory performance to classify high fracture risk patients. The proportion of the study population who would be referred

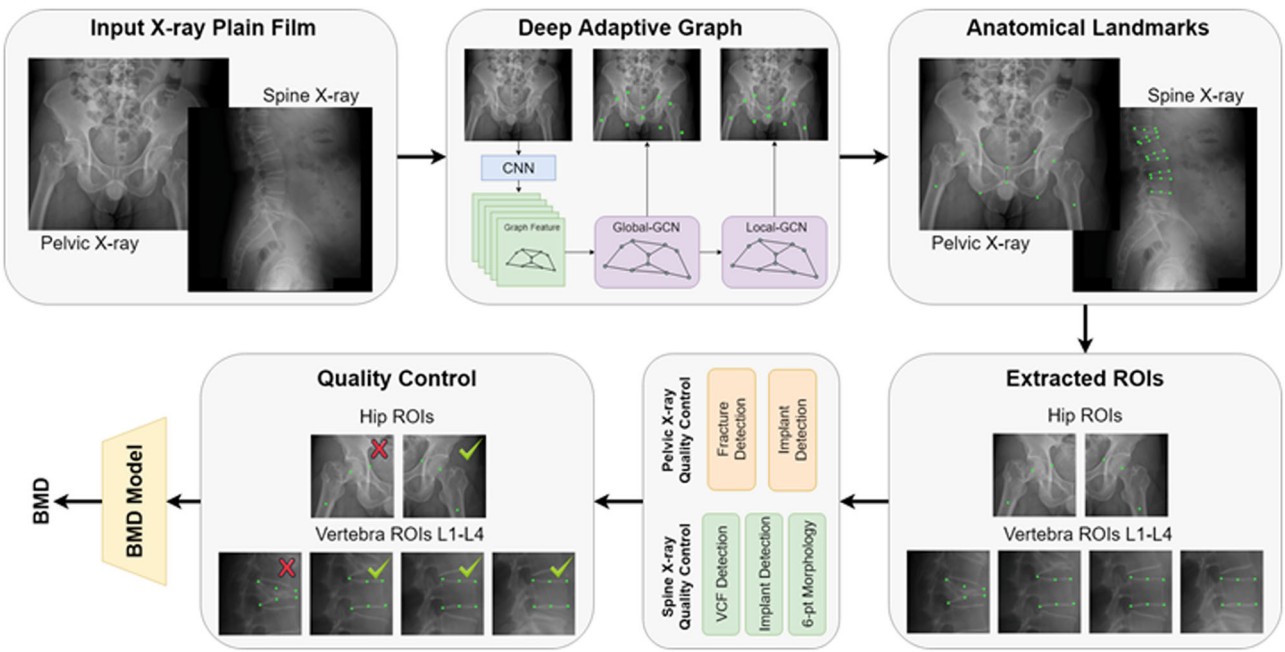

**Fig. 1** Schematic representation of the workflow for hip and spine BMD estimation.

**Table 1 Patient characteristics of the study population.**

| | Hip testing set | | | Spine testing set | | |
|---|---|---|---|---|---|---|
| | Hologic training | Hologic testing | p*** | Hologic training | Hologic testing | p*** |
| Number | 5633 | 5164 | | 7307 | 18,175 | |
| Female, n (%) | 4380 (77.8) | 3997 (77.4) | 0.66 | 5815 (79.6) | 14,469 (79.6) | 0.96 |
| Mean age (sd), years | 72.8 (11.0) | 72.2 (11.2) | 0.004 | 67.2 (12.5) | 67.1 (10.6) | <0.001 |
| Median time (IQR) between DXA and radiographs | 31 (8, 81) | 29 (8, 77) | 0.13 | 24 (22, 27) | 16 (5, 48) | <0.001 |
| Mean BMI (sd), kg/m$^2$* | 23.8 (3.9) | 23.9 (3.9) | 0.08 | 24.6 (3.8) | 24.5 (3.8) | 0.06 |
| Mean BMD (sd) g/cm$^2$ | 0.678 (0.159) | 0.689 (0.156) | <0.001 | 0.740 (0.173)** | 0.762 (0.176)** | <0.001 |
| Median T-score (IQR) | −1.6 (−2.4, −0.6) | −1.5 (−2.3, −0.6) | <0.001 | −2.3 (−3.3, −1.3)** | −2.2 (−3.1, −1.1)** | <0.001 |
| Osteoporosis, n (%) | 1356 (24.1) | 1110 (21.5) | <0.001 | 3291 (45.0)** | 7860 (43.3)** | 0.009 |

*BMI was not available in 330/66 patients in the hip training/testing sets and 223 and 623 in the spine training/testing sets.
**Calculated based on vertebrae with the lowest BMD.
***Categorical variables (gender and osteoporosis) were compared using Chi-square test. Means were compared using student t-test and medians were compared using Wilcoxon rank-sum test. Two-sided p values were reported.

for DXA was 11.8% in the hip cohort and 29.6% in the spine cohort.

**External validation.** We identified 2060 patients with paired GE DXA-pelvis radiographs and 3346 patients with paired GE DXA-lumbar spine radiographs (Supplementary Table 2). The GE BMD values were converted to Hologic values using the manufacturer-provided equations (Supplementary table 3). Supplementary Table 4 summarizes the model performance by comparing model-predicted BMD and GE DXA-measured BMD and Supplementary Table 5 summarized the discriminatory performance. The Pearson's correlation coefficients between GE DXA-measured and model-predicted BMD were 0.90 for the hip and 0.89 for the hip and lumbar spine (Supplementary Fig. 4). The model remains robust with good linear correlation, calibration and minimal bias across different age and sex strata. The discriminatory performance is also excellent, with an AUPRC of 0.87 for the hip and 0.89 for the spine model. We further test our tool using 34 pairs of GE DXA-hip radiographs and 179 pairs of DXA-lumbar spine radiographs from the Wuhan Hospital of

Traditional Chinese Medicine. The Pearson correlation coefficient was 0.93 for the hip model and 0.86 for the spine model.

**Real-world experiment.** Next, we implemented the tools in the central inference platform connected to the picture archiving and communication system (PACS) in the Chang Gung Memorial Hospital (CGMH, Linkou branch) to study the real impact of our tool to screen osteoporosis. The hospital PACS relayed all newly acquired images to the inference platform daily. In total, 2388 consecutive pelvis (1858 patients, 43.2% women) and 9741 lumbar spine radiographs (5336 patients, 40.8% women) in those aged 40–90 years were conducted between January and May 2021. The tool excluded 816 pelvis radiographs and 1715 spine radiographs due to poor image quality, inappropriate positions, implants, and fractures that may impede BMD estimation. The percentages of images passing through the entire pipeline and successfully reporting a predicted BMD were 79.0% for pelvis radiographs and 82.3% for spine radiographs. Among these, 5206 (84.8%) patients with hip or spine radiographs were classified or excluded as osteoporotic with high PPV or NPV for osteoporosis using thresholds reported in Table 4. Finally, only 933 (15.2%)

**Table 2 Summary of performance metrics of predictive models for Hologic BMD.**

| Patient strata | Number of ROIs | Predicted vs. measured mean BMD (sd, g/cm²); p** | Correlation coefficient | Linear regression R², RMSE | Calibration slop, CITL | Bland-Altman bias (g/cm²; sd) |
|---|---|---|---|---|---|---|
| *The hip testing set (Hologic)* | | | | | | |
| Overall | 5164 | 0.692 (0.144) vs. 0.689 (0.156); p < 0.001 | 0.92 | 0.84, 0.062 | 0.982, −0.003 | −0.003 (0.062) |
| Female | 3997 | 0.668 (0.137) vs. 0.661 (0.144); p < 0.001 | 0.91 | 0.83, 0.056 | 0.961, −0.007 | −0.007 (0.059) |
| Male | 1167 | 0.774 (0.137) vs. 0.782 (0.151); p < 0.001 | 0.89 | 0.80, 0.062 | 0.985, 0.008 | 0.008 (0.068) |
| 40–59 years | 712 | 0.796 (0.132) vs. 0.790 (0.140); p = 0.010 | 0.91 | 0.82, 0.056 | 0.987, −0.006 | −0.006 (0.061) |
| 60–74 years | 1817 | 0.722 (0.132) vs. 0.717 (0.141); p < 0.001 | 0.90 | 0.82, 0.057 | 0.963, −0.005 | −0.005 (0.061) |
| 75–90 years | 2635 | 0.644 (0.135) vs. 0.642 (0.148); p = 0.175 | 0.91 | 0.82, 0.057 | 0.998, −0.002 | −0.002 (0.062) |
| *The spine testing set (Hologic)** | | | | | | |
| Overall | 57,662 | 0.837 (0.172) vs. 0.839 (0.186); p < 0.001 | 0.90 | 0.81, 0.081 | 0.978, 0.003 | 0.003 (0.081) |
| Female | 46,349 | 0.813 (0.162) vs. 0.813 (0.176); p < 0.001 | 0.89 | 0.80, 0.079 | 0.969, 0.000 | 0.000 (0.079) |
| Male | 11,313 | 0.931 (0.177) vs. 0.945 (0.191); p = 0.94 | 0.89 | 0.79, 0.088 | 0.958, 0.014 | 0.014 (0.088) |
| 40–59 years | 14,501 | 0.912 (0.160) vs. 0.909 (0.174); p < 0.001 | 0.90 | 0.80, 0.077 | 0.973, −0.003 | −0.003 (0.081) |
| 60–74 years | 26,935 | 0.827 (0.166) vs. 0.828 (0.181); p = 0.003 | 0.90 | 0.80, 0.079 | 0.978, 0.001 | 0.001 (0.079) |
| 75–90 years | 16,226 | 0.784 (0.166) vs. 0.795 (0.188); p < 0.001 | 0.88 | 0.79, 0.086 | 1.004, 0.011 | 0.011 (0.086) |
| L1 | 12,731 | 0.742 (0.150) vs. 0.747 (0.167); p < 0.001 | 0.87 | 0.76, 0.081 | 0.968, 0.006 | 0.006 (0.082) |
| L2 | 15,809 | 0.816 (0.160) vs. 0.821 (0.177); p < 0.001 | 0.90 | 0.81, 0.078 | 0.993, 0.006 | 0.006 (0.078) |
| L3 | 15,679 | 0.873 (0.163) vs. 0.873 (0.181); p = 0.22 | 0.90 | 0.80, 0.080 | 0.993, 0.001 | 0.001 (0.080) |
| L4 | 13,443 | 0.909 (0.167) vs. 0.907 (0.182); p = 0.01 | 0.88 | 0.78, 0.084 | 0.966, −0.002 | −0.002 (0.085) |

*Calculated per eligible vertebrae.
**Means were compared using student *t*-test and medians were compared using Wilcoxon rank-sum test. Two-sided *p* values were reported.

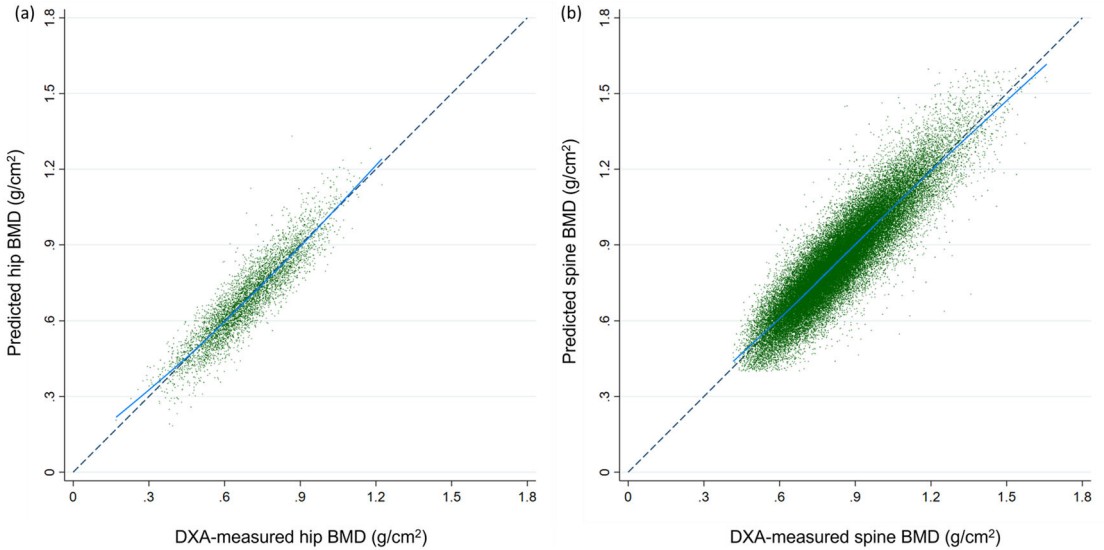

**Fig. 2 The calibration plots for predicted-measured BMD.** The calibration plots show predicted BMD values against DXA-measured BMD values for assessment of model performance. **a** Five thousand one hundred and sixty-four pairs of predicted-measured hip BMD (5164 patients), and **b** 57,662 pairs of predicted-measured lumbar vertebral BMD (18,175 patients). Each point represents a data pair of predicted and measure BMD. The points close to the diagonal line suggests good calibration.

**Table 3 Discriminatory performance (%) of the predicted BMD to classify hip/lumbar vertebral osteoporosis and high-risk groups for major osteoporotic or hip fractures.**

| Discriminatory measures | Hip osteoporosis (T-score ≤ −2.5 | Lumbar vertebral osteoporosis (vertebrae with the lowest T-score ≤2.5) | 10-year risk of major osteoporotic fracture ≥20% | 10-year risk of hip fracture ≥3% |
|---|---|---|---|---|
| Number of patients, % | 1110, 21.5 | 7860, 43.3 | 530, 10.4 | 2254, 44.2 |
| OR (95% CI) | 74.80 (61.05–91.65) | 38.22 (35.12–41.59) | 107.92 (81.98–142.06) | 79.29 (66.03–95.24) |
| AUROC/AUPRC | 0.97/0.89 | 0.92/0.89 | 0.97/0.83 | 0.97/0.96 |
| Accuracy (%; 95% CI) | 91.7 (90.9–92.5) | 86.2 (85.7–86.7) | 95.0 (94.3–95.6) | 90.0 (89.1–90.8) |
| Sensitivity (%; 95% CI) | 80.2 (77.7–82.5) | 83.5 (82.7–84.3) | 69.6 (65.5–73.5) | 88.2 (86.8–89.5) |
| Specificity (%; 95% CI) | 94.9 (94.1–95.5) | 88.3 (87.7–88.9) | 97.9 (97.5–98.3) | 91.4 (90.3–92.4) |
| PPV (%; 95% CI) | 81.1 (78.9–83.1) | 84.5 (83.8–85.2) | 79.5 (76.0–82.7) | 89.1 (87.8–90.2) |
| NPV (%; 95% CI) | 94.6 (94.0–95.2) | 87.6 (87.0–88.0) | 96.5 (96.1–96.9) | 90.7 (89.7–91.6) |

patients were advised to take the DXA examination (Supplementary Fig. 5). At the same period, 3008 DXA examinations were conducted in CGMH, Linkou branch.

## Discussion

Osteoporosis is a silent disease before fragility fractures, leading to multiple morbidities and increased mortality in affected patients[4]. Therefore, population-based screening is imperative for identifying at-risk patients and implementing preventive measures. DXA is the preferred screening modality to screen osteoporosis but is of limited availability, especially for the developing counties[8–13] and underutilized in the well-resourced area such as the US[15]. In addition to improving DXA availability and utilization, opportunistic osteoporosis screening using other imaging modalities is a potential strategy to expand screening populations. In CGMH, approximately 80% of patients aged 40–90 years with pelvis or spine radiographs had not been screened by DXA previously. Our automated, reliable tool can evaluate fracture risk using these radiographs "already" conducted for other indications to identify at-risk patients, who are not screened by DXA without additional cost, time, and radiation.

The performance of the tool is robust with DXA as a reference and compared favorably with other opportunistic osteoporosis

screening tools based on CT attenuation of the spine (area under the receiver-operator curve [AUROC], 0.83)[22] and machine-learning-based T-score simulation (accuracy, 82%)[23]. In comparison, our tool correlated well with gold standard Hologic or GE DXA-measured BMD in both internal and external testing sets with excellent discriminatory performance to classify hip and spine osteoporosis (AUROC, 0.97 and 0.92, respectively) and stratify patient fracture risks. Clinical testing of our automated tool in consecutive patients with pelvis or spine radiographs found that approximately 80% of them could be automatically screened for osteoporosis. Among them, our tool classified osteoporosis with excellent PPV or NPV for osteoporosis for 5206 patients who were mostly not examined by DXA; during the same period, 3008 DXA examinations were conducted. The real-world evidence demonstrated that our automated tool could expand opportunistic screening to a broader population at risk.

BMD is not the only determinant of fracture risk[32]. A history of osteoporotic fracture is one of the clinical risk factors for FRAX but often are unnoticed because many patients with occult hip fractures and VCFs are asymptomatic[22,33]. We exploited the excellent spatial resolution of radiographs to identify hip implants and unsuspected fragility fractures before estimation of BMD. The tool incorporates our previously published PelviXNet[34] to

**Table 4 Osteoporosis classification results at 95% PPV and 95% NPV thresholds on Hologic testing data.**

| Threshold with high PPV ≥95% | Hip | Spine |
|---|---|---|
| Threshold BMD (g/cm², T-score (T1) | 0.513, −2.9 | −3.0* |
| *PPV for osteoporosis* | | |
| TP/(TP + FP), n/N | 524/549 | 4750/5030 |
| PPV (95% CI) | 95.5 (93.5–96.8) | 94.4 (93.8–95.0) |
| PPV for 10-year major fracture risk ≥20% | | NA |
| TP/(TP + FP), n/N | 230/283 | |
| PPV (%, 95% CI) | 81.3 (76.9–85.0) | |
| PPV for 10-year hip fracture risk ≥3% | | NA |
| TP/(TP + FP), n/N | 492/498 | |
| PPV (95% CI) | 98.8 (97.4–99.5) | |
| Threshold with high NPV ≥95% | | |
| Threshold BMD (g/cm², T-score (T2) | 0.580, −2.3 | −2.1* |
| *NPV for osteoporosis* | | |
| TN/(TN + FN), n/N | 3812/4008 | 7418/7765 |
| NPV (95% CI) | 95.1 (94.5–95.6) | 95.5 (95.1–96.0) |
| NPV for 10-year major fracture risk ≥20% | | NA |
| TN/(TN + FN), n/N | 3884/3953 | |
| NPV (95% CI) | 98.3 (97.8–98.6) | |
| NPV for 10-year hip fracture risk ≥3% | | NA |
| TN/(TN + FN), n/N | 2575/2839 | |
| NPV (95% CI) | 90.7 (89.8–91.5) | |
| *Prediction categorization* | | |
| Patient number (%) | | |
| Predicted BMD or T-score <T1 | 549 (10.6) | 5030 (27.7) |
| T1 ≤ predicted BMD or T-score <T2 | 607 (11.8) | 5380 (29.6) |
| Predicted BMD or T-score ≥T2 | 4008 (77.6) | 7765 (42.7) |

*FN false negative, FP false positive, NA not available, NPV negative predictive value, PPV positive predictive value, TN true negative, TP true positive.*
*\*Based on the vertebrae with the lowest T-score.*

detect hip fracture and newly developed algorithms to detect hip implants. Furthermore, we developed a VCF assessment algorithm based on a Deep Adaptive Graph network (DAG)[35], which determines anatomical landmarks for standard six-point vertebral morphometry that facilitates VCF detection using the widely accepted semiquantitative Genant visual method[36,37]. Therefore, our tool could evaluate fragility fracture risk based on a single radiograph. However, other patient-related clinical risk factors (e.g., comorbidity, medication, and lifestyle) require input from electronic medical records.

Opportunistic osteoporosis screening using other imaging modalities has been reported previously, but none had been clinically examined as comprehensive as our study. The best-studied strategy is the use of abdominal CT to predict spine BMD[19,20,23], classify osteoporosis based on CT attenuation[22], simulated BMD[19,20], T-score[23], or detect osteoporotic fractures[38]; or use imaging biomarkers to predict the risk of fractures[24]. Julien Smets et al. reviewed machine learning solutions for osteoporosis[39]. Among five studies using CT scans to predict BMD, the best correlation coefficient between estimated and CT-simulated spine BMD was 0.94[21]. An earlier study compared the CT Hounsfield units over a manually annotated ROI involving vertebral body trabecular bone with its paired DXA T-score; this approach for detection of osteoporosis yielded an AUROC of 0.83[22]. Deep learning-based models provided a better correlation between predicted and reference values, but were only validated in small datasets[19,20,23]. A larger study testing the performance of simulated T-scores on a larger dataset of 1843 CT-DXA pairs achieved an accuracy of 82% to detect osteoporosis[23]. This algorithm was integrated with VCF identification and CT trabecular density as biomarkers, and its performance for the prediction of 5-year fracture risks was compared favorably with the performance of FRAX-NB[24]. Osteoporosis and fragility fracture risk have also been assessed on dental[40,41], hip[42,43], and spine radiographs[41,44], and magnetic resonance imaging[45]. However, only three were validated against standard DXA-based hip or spine BMD. The best AUROC was 0.92 for hip[42] and 0.73 for spine osteoporosis classification using small testing sets (131 and 345 patients, respectively)[44]. These studies demonstrated the feasibility of using non-DXA modalities to screen osteoporosis, although the applicability and usability of such tools in real clinical settings are questionable.

In contrast, the present study provided a fully automated tool enabling opportunistic screening for osteoporosis and evaluating fragility fracture risk using plain radiographs of the hip and spine. Our tool utilizes ubiquitous, low-cost radiographs that involve substantially lower radiation exposure than CT-based tools to assess both the hip rather than the spine alone (e.g., using CT-based tools). Furthermore, we envision that other musculoskeletal radiographs may also be used to predict bone density and fracture risks, regardless of the original purpose of the images. This strategy used radiographs already taken for other indications, therefore requiring no additional patient time or radiation exposure with minimal costs but may substantially improve the risk profiling for fragility fractures.

This study had several limitations. First, CGMH is a medical center in which the patients tend to have more severe diseases. A large proportion of patients have fractures or implants. Our study population may not have represented the healthier population. However, because the tool was developed based on the more complex population, the ROI localization, quality check, and BMD prediction processes can presumably be readily adapted to populations with fewer complications. In addition, the performance of our tool remains robust when testing on external data. Second, the calculation of FRAX in this study did not consider past medical history, medication use, family medical history, alcohol consumption, and smoking status because this information requires input from the hospital information system. However, the performance assessment should not vary much because these parameters are identical for FRAX based on the DXA-measured or model-predicted BMD. The clinical implementation of the tool can report full FRAX results when digital data are available. Third, the tool was created using the reference BMD values reported by Hologic DXA scanners alone. However, the model's performance remains robust in a test set of paired GE DXA and plain radiographs and external sources. The GE BMD measurements were converted to corresponding Hologic BMD values using the algorithm provided by the Hologic manufacturer. It seems the conversion has a small negative effect on model performance. A specific model for GE DXA is needed to maximize performance. Fourth, the performance of the prediction tool is influenced by the radiograph image quality. In addition to existing fractures, accurate BMD prediction may be impeded by foreign bodies, implants, bowel gas, and bone pathologies (e.g., avascular necrosis or severe osteoarthritis). The actual rate of radiographs that could be evaluated for BMD and fracture risk is around 80% in our real-world test. Depending on a patient's specific indications, radiographs are often examined repeatedly.

Therefore, the per-patient success rate will potentially increase as more radiographs become available over time.

This study demonstrated that a robust opportunistic screening tool for osteoporosis and fracture risk assessment, based on conventional radiographs obtained for various indications, provided VCF detection, BMD, and fracture risk estimation in a fully automated process. This tool leveraged state-of-the-art deep learning algorithms to provide an efficient strategy for population-based opportunistic screening with minimal additional cost. Integrating this automated tool into the hospital information system may expand osteoporosis screening to a broader population at risk.

## Methods

**Setting**. This study was approved by the Institutional Review Board at the CGMH and was conducted in accordance with the tenets of the Declaration of Helsinki. This study was approved by the Institutional Review Board at the CGMH (approval number: 202000254B0, 202100346B0, 202101180B0). The requirement for informed consent was waived because the data used in this paper were fully de-identified to protect patient confidentiality. This study was performed using data from CGMH, the largest private hospital system in Taiwan, which includes seven acute hospitals with 10,050 beds that received 8.2 million outpatient visits and 2.4 million inpatient care visits. The study was conducted in collaboration between the CGMH and PAII Inc., a research subsidiary of Ping-An Technology that focuses on state-of-the-art computer vision algorithm development. PAII Inc. used clinical images and clinical data from CGMH to create automated BMD and fracture risk estimation tools. The provided data were fully encrypted to prevent patient confidentiality leaks. Except for the training and validation image data, PAII Inc. remained blinded to other clinical and testing datasets.

The study population consisted of 184,339 patients with at least one central DXA from January 2006 to December 2020 and were aged 40–90 years on the DXA index date. The study population was also required to have adequate radiographs of the pelvis or lumbar spine within 180 days from the index date. For patients with multiple DXA and plain film radiographs, the earliest pair was used. We performed a quality check for plain films to ensure that these images were suitable for BMD prediction; after the exclusion of inadequate plain films, model building and testing were performed based on a cohort of 10,797 patients with at least one Hologic DXA-pelvis radiograph pair and 25,482 patients with at least one lateral radiograph of the lumbar spine–DXA pair (Supplementary Fig. 1). The patients were randomly allocated into the training and testing set by simple random sampling in which each patient has an equal probability of selection, and sampling is without replacement. Patients with GE DXA-plain film pairs were used as the separate testing sets (hip testing set, $n = 2060$; spine testing set, $n = 3346$). We also include 34 pairs of GE DXA-hip radiographs and 179 pairs of DXA-lumbar spine radiographs from the Wuhan Hospital of Traditional Chinese Medicine to do external validation.

We also tested the algorithms in a clinical setting to ascertain the number and proportions of patients with hip or spine radiographs who may benefit from the tool. The algorithms were packaged in docker containers and implemented on the PACS-linked inference platform of CGMH, based on the Nvidia Triton architecture. We tested the model using consecutive radiographs conducted between January 2021 and May 2021.

**BMD measurement**. Proximal femoral and lumbar spine DXA scans were performed using a Hologic QDR-4500A fan-beam densitometer (Bedford, MA, USA) during 2005–2010 and a Hologic Discovery model A densitometer during the period 2011–2021. The GE DXA scanner was the Lunar iDXA system (Madison, WI). The scans were analyzed following recommendations issued by the Taiwan Radiological Society[46], amended from the International Society for Clinical Densitometry, ISCD (Supplementary methods)[47]. Hip $T$-scores were calculated using the revised NHANES III white female reference values[48,49]. Because there is no international reference standard for the lumbar spine BMD, lumbar $T$-scores were calculated using the manufacturer's reference values. For each patient, the lowest $T$-score of the hip or lumbar vertebrae was used to categorize osteoporosis or calculate FRAX risk.

**Acquisition and preprocessing of radiographs**. The radiographs were collected from the PACS and anonymized before the study procedure. Most radiographs were produced using the Canon CDXI 710C (82.5% for the hip and 86% for the spine). The peak kilovoltage (kVp) range is mainly 70–80 kV for the hip and 90–95 kV for the lumbar spine. No performance difference was observed between different machines or kVp (Supplementary Table 6). The images were converted to grayscale and resized to a resolution of 0.15 mm × 0.15 mm pixel spacing, then stored as 12-bit images. A deep adaptive graph (DAG) landmark detection method was developed to formulate the anatomical landmarks of the pelvis and spine as graphs and to robustly and accurately detect these landmarks[35]. We detected 16 anatomical landmarks on hip radiographs, including 12 landmarks on the pelvic

boundary and four landmarks on the femoral head and trochanter. We detected six anatomical landmarks for each of the lumbar vertebrae on spine radiographs from L1 to L4. ROIs were extracted from the radiographs and used as input for the BMD prediction model based on the detected anatomical landmarks. For hip radiographs, ROIs of the left and right hips were extracted. For the lumbar spine, ROIs were extracted for each vertebra from L1 to L4. The ROIs were used as input for the BMD prediction model. A schematic representation of the pipeline, models and examples of the detected anatomical landmarks and ROIs used to predict BMD is shown in Fig. 1.

**Anatomical landmark detection via deep adaptive graph (DAG)**. The anatomical landmarks were detected using DAG, a method introduced in our previous publication[35]. The details of DAG were described in Supplementary methods. In our experiment, the hip and spine DAG models were trained using 3306 pelvic radiographs and 1076 spine radiographs with expert annotations, respectively. The radiographs used to train the DAG models are excluded from the test sets used to evaluate the BMD estimation models. The DAG models are evaluated on 876 pelvic and 290 spine radiographs and report 4.29 + −3.29 mm and 1.22 + −3.23 mm localization errors.

**Automated radiograph quality assessment procedure**. Some medical conditions may affect the hip and vertebra anatomy, making plain films unsuitable for BMD estimation. The most common conditions include implantation and fracture. Therefore, we conducted an automated quality assessment to exclude hips and vertebrae with implants or fractures unsuitable for BMD prediction.

**Quality assessment of hip radiographs**. We detect hip fracture and implant (joint prosthesis, screws, plates, or cement) in the quality assessment process and exclude them from the downstream BMD estimation. An existing model, PelviXNet[34], is used to detect the hip fracture. PelviXNet consists of a DensetNet-121 backbone neural network and a Feature Pyramid Network and was trained on 5204 pelvic radiographs that had been annotated by experienced physicians using an efficient and flexible point-based annotation scheme. In addition to detecting hip fracture, we trained another network with an identical structure to PelviXNet using 2973 pelvic radiographs to detect implants. The maximum responses of the fracture and implant detection networks in the hip ROI are calculated as the classification scores for hip fracture and implant, respectively. The fracture detection model, PelviXNet, was evaluated on 1888 pelvic radiographs covering various medical conditions (e.g., implants and periprosthetic fracture) and reports 92.4% sensitivity and 90.8% specificity. The implant detection model was evaluated on 715 randomly selected pelvic radiographs and reports 99.9% sensitivity and 99.7% specificity.

**Quality assessment of spine radiographs**. The adult official positions of the ISCD advise excluding vertebrae that are abnormal and non-assessable or have a more than a 1.0 $T$-score difference between the vertebra in question and adjacent vertebrae[50]. Therefore, the automated quality assessment procedure for spine radiographs is performed in three steps: implant and VCF detection, six-point morphology analysis and assessment for $T$-score of nearby vertebrae. The implant/VCF detection model had the same architecture as PelviXNet and was trained on 1485 expert-annotated lateral spine radiographs to produce probability maps for implant and VCF. The L1 to L4 vertebrae were classified as normal, VCF, and implant by the annotator. A supervision mask was then generated by filling the vertebra polygons produced by DAG using the annotated label.

Using the predicted implant and VCF probability maps, the maximum responses in the vertebrae polygons were regarded as the classification scores. Vertebrae with a positive implant or VCF detection results were excluded, and the remaining vertebrae were analyzed by six-point morphology. Specifically, six landmarks were detected for each vertebra, including two anterior points, two posterior points, and two middle points of the top and bottom vertebral plates. Four distances were calculated from these six points: anterior height $h_a$, posterior height $h_p$, middle height $h_m$, and vertebra width $w$. The three heights were calculated as the pairwise distances between the two anterior, posterior, and middle points. The vertebra width was calculated as the mean distance between the anterior and posterior points. Three criteria were used to identify vertebrae with abnormal deformity, following the widely accepted Genant visual semiquantitative method[37], with modifications to facilitate automated measurement and fracture detection:

$$\frac{\min(h_a, h_p)}{\max(h_a, h_p)} < 0.8, \quad (1)$$

$$\frac{h_m}{\max(h_a, h_p)} < 0.6, \quad (2)$$

$$\frac{\max(h_a, h_p)}{w} < 0.55. \quad (3)$$

The first criterion aimed to detect wedge and crush fractures, where the anterior and posterior heights were reduced. The second criterion aimed to detect a

biconcave fracture, where the middle height was reduced. The last criterion aimed to detect severe VCF cases where the overall height of the vertebra was significantly reduced. If a vertebra met any of the three criteria, it was considered abnormal and excluded from downstream processing. These criteria only detected moderate to severe compression fractures to avoid ambiguity in determining mild or borderline deformities. The vertebrae with more than one standard deviation difference from their neighbors were excluded from the analysis. We comply with the ISCD positions that only those with two or more assessable vertebrae were included for analysis[50].

To evaluate the performance of the spine radiograph QA module, we randomly selected 200 spine radiographs from the test set and manually labeled implant and VCF. The implant and VCF detection module report 91.5% and 93.2% sensitivity and 99.5% and 91.5% specificity. Some mild VCFs are not detected by the VCF detection module alone.

**Algorithm development and training procedure for BMD prediction**. We developed a deep learning algorithm to estimate the hip/spine BMD from each corresponding ROI. The neural network employs a backbone network to encode the input ROI as a feature vector and two consecutive fully connected layers with ReLU activation functions to produce the estimated BMD. We evaluated multiple backbone networks (i.e., VGG-11, VGG-16, ResNet-18, ResNet-34) in earlier experiments and empirically found that VGG-16 and ResNet-34 produce the best BMD prediction results for spine and hip BMD prediction, respectively. The model using only image-based features already performs strongly, and the addition of age and gender does not improve the model's performance. Therefore, we choose to use the VGG16 backbone without age and sex in later model development (Supplementary Tables 7 and 8). L1–L4 vertebrae have slightly different geometries and distinct BMD statistics; therefore, the vertebra index information was required by the model to predict the BMD accurately. In the spine model, the vertebra index (from L1 to L4), encoded by a one-hot vector of length 4, appended to the feature vector in the neural network before the last fully connected layer. During training, ROIs were augmented by random affine transformation and subsequently resized to $512 \times 512$ pixels. The L1 distance between the predicted BMD and the ground truth BMD obtained from DXA was regarded as the training loss. A fourfold cross-validation procedure was conducted, and ensemble learning was adopted to combine the predictions of the four trained models during inference. Pseudocodes for BMD estimation are provided in the Supplementary Tables 9 and 10.

**Evaluation of BMD prediction performance**. Evaluation of all performance measures was performed only on the test datasets. The Bland–Altman plot visualized the agreement between predicted and measured BMD scores, and Pearson's correlation coefficient was calculated. The tool's calibration was evaluated by comparing the mean risk calculated based on predicted BMD and the mean risk based on DXA-measured BMD. The following measures were calculated to evaluate the overall calibration: calibration slope and calibration-in-the-large. Osteoporosis results were considered positive when $T$-score $\leq -2.5$. Ten-year probabilities of major fracture and hip fracture with total hip BMD were calculated for each patient using the FRAX tool with risk estimators specific to the Taiwanese population (https://www.sheffield.ac.uk/FRAX/; FRAX Desktop Multi-Patient Entry, version 4.0). The FRAX parameters used in this study include age, sex, weight, height, and BMD. FRAX risks with and without BMD were calculated separately. For each patient, the lowest BMD was used to calculate the $T$-score and FRAX risk. Ten-year risk scores of ≥3% for hip fracture and ≥20% for major osteoporotic fracture were considered high-risk, based on the intervention threshold established in the Taiwan Osteoporosis Practice Guidelines[51] and the recommendations of the National Osteoporosis Foundation[52]. The overall discriminative abilities to discern osteoporosis and high-risk patients were evaluated using the AUROC and AUPRC. Other measures were also calculated, including sensitivity, specificity, positive predictive value, and negative predictive value. Two-sided $p$ values were reported throughout the manuscript. Analyses were conducted using Stata software, version 16.1 (StataCorp, College Station, TX, USA).

**Reporting summary**. Further information on research design is available in the Nature Research Reporting Summary linked to this article.

## Data availability

The sample testing imaging data generated in this study have been deposited in the public repository (https://doi.org/10.5281/zenodo.5216219). The full original imaging data are available under restricted access for the policy of the Chang Gung Memorial Hospital and data privacy laws. Researchers who are interested in our work can request access to the de-identified raw images for academic purposes. The request should be made to the corresponding author and the access will be granted within a month. Use of data is limited to research purposes and redistribution of data is not allowed.

## Code availability

The code used to train and evaluate the model performance is not openly available due to the use of proprietary software packages and infrastructures including a general deep learning model training and evaluation platform that is used not only for the described

study but also across other projects at PAII Lab. Pseudocodes are provided in the Supplementary Tables 9 and 10. We provided a Gigantum project (https://gigantum.com/xraybmd/nc-bmd-cpu) to test our model. The instructions for the Gigantum project are provided at the end of the supplementary materials. The inference services are available from the corresponding author upon request. Researchers who are interested in our work can request access to the Gigantum project or inference services for academic purposes. The request should be made to the corresponding author and the access will be granted within a month.

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

## Acknowledgements
The authors thank the funding from Chang Gung Memorial Hospital (CLRPG3H0012, CLRPG3H0013, CORPG3J0191) and the Ministry of Science and Technology (MOST 109-2321-B-182A-007-) for supporting the research. The authors would like to thank Ms. Meng-Jiun Chiou and Yu-Ying Chen for data preparation and statistical analysis.

## Author contributions
C.I.H., K.Z., C.H.L. contributed equally to the manuscript. C.F.K., C.I.H., S.M. and L.L. present the conception of the work; C.F.K., S.M. and L.L. designed the study; C.F.K., L.L., G.T.X. and J.X. obtained funding to support the work and built up the research team for joint development; C.F.K., C.I.H., C.H.L., F.P.C. and L.M. acquired the data; K.Z., W.J.L., Y.R.W., X.Y.W., F.K.W. labeled the images for further analysis; S.M., K.Z., W.J.L., Y.R.W., X.Y.W. and F.K.W developed the deep learning model; C.F.K., J.X., S.M. and L.L. conducted the statistical analysis; S.M. created the new software for labeling image; C.H.L. prepared the real-world inference system and conducted the experiments; C.F.K., C.I.H., C.H.L. and S.M. drafted the manuscript; C.F.K., L.L., C.H.L., G.T.X. and J.X.H. substantively revised it. All the authors made substantial contributions to this work and have critically reviewed the manuscript before submission.

## Competing interests
The authors declare no competing interests.
