## [Peer Review File · Nature Communications]

Reviewers' Comments:

Reviewer #1:

Remarks to the Author:

The study presents a deep-learning based tool to estimate BMD from pelvis radiographs as well as sagittal radiographs of the lumbar spine, which has been packaged together with a simplified version of the FRAX scoring system to predict the risk of osteoporotic fracture. The methods are solid and the deep learning models have been trained on relatively large datasets. The paper is professionally written and presented, and is enjoyable to read.

However, I have a major criticism; the motivation of the study is clear but I do not think that it is valuable enough to grant publication on Nature Communications. The novel method "replaces" DXA for the calculation of the BMD, which is interesting from an academic point of view, and the opportunistic use of radiographs is clinically relevant to some extent; however, I do not see a major clinical advantage in doing that. DXA is cheap, fast, widely available and has a lower X-ray dose in comparison with standard radiographs. I understand the point of the Authors about the fact that "current DXA-based programs screen fewer than one-third of eligible women and one-tenth of eligible men", but I do not see how this makes "osteoporosis screening based on DXA inadequate". Obtaining plain radiographs from the general population (other than those needing them for clinical purposes) would be even more difficult, and ethically questionable due to the additional dose of ionizing radiation. DXA remains the golden standard for osteoporosis screening; plain radiographs will not replace DXA in this respect. If a patient needs an investigation of the bone quality, he/she should undergo DXA (or qCT).

Besides, since the models are trained to estimate the BMD based on DXA-measured values, they cannot be "better" than DXA, which has its own limitations. If the models were trained using qCT as ground truth, then I would potentially see an advantage since qCT provides "higher quality" data with respect to both BMD and plain radiographs and it would allow go beyond the state-of-the-art, but this is not the case here.

Other comments:

- as far as I know, Nature Communications requires peer review of the computer code, but I could not find out how to access the source code. Full access to it should be provided (neural network architecture, training and inference).

- Abstract: since the main tool discussed in the paper performs a prediction of the BMD value, please report data about the error in its estimation rather than (or in addition to) the accuracy in the identification of osteoporosis/fracture risk. The calculation of the fracture risk is indeed based on the FRAX method, although with modifications and simplifications; its accuracy is therefore driven by the accuracy in estimating the BMD.

Reviewer #2:

Remarks to the Author:

This manuscript presents an opportunistic screening tool for osteoporotic fractures in hip and lumbar spine using plain radiographs. The tool has high clinical value since it runs automatically in the background and can identify high risk osteoporotic patients and measure BMD during their routine hospital check-up. It can generate big cost saving for the patients and the hospitals. The tool was trained and tested on a large data set. It was packaged and integrated in a PACS system and was the first to be evaluated in a large trial in real clinical setup. They developed state-of-the-art AI techniques including deep adaptive graph for landmark detection, DenseNet for quality assessment in the tool. The models were robust under different radiograph image quality and achieved high performance. The manuscript is well organized and clearly written.

Detailed comments:

1) It would be desirable to have more in-depth reviews on computational methods for BMD measurement and fracture detection, including deep learning and traditional image analysis

methods.

- 2) Why exclude data obtained using GE DXA scanner?
- 3) How do you split the training and test sets?
- 4) Line 98, please specify what unsuitable vertebrae are. Since 30% of vertebrae are excluded, is the quality assessment automatic? What will happen to those excluded vertebrae? A warning will be issued?
- 5) Line 99, how were BMD measured? By how many operators? Is there inter-operator variability?
- 6) Line 151, among the 90.4% of radiograph with predicted BMD, do you assess the accuracy of BMD value? Do you calculate the distribution of BMD? Is it in line with those in the training set?
- 7) Are all the radiographs acquired from the same machine? If not, any performance difference?
- 8) Line 381, is the data used to train DAG models part of the training set listed in Table 1? If not, please specify.
- 9) Line 444, does the DXA-based BMD measurement consider age and sex?
- 10) Line 450, how to label vertebra index? How accurate?
- 11) Since all data are from a single hospital center. How good would this model be generalized for data from a new center? Do you need retraining or transfer learning? Please discuss.

Reviewer #3:

Remarks to the Author:

This work describes a fully automated system using age and sex information in addition to radiographic images of the hip or spine to estimate BMD and infer risk via osteoporosis and FRAX measurements. The document is well written and is on average clearly detailed. Achieved performances were highly promising and would be of great impact in the high risk population screening. However, several points concerning performance benchmarking, results reporting and metrics used remains to be addressed. Overall, the results should be more mitigated by the serious limitations observed, particularly regarding the lack of intermediate validation, which leaves room for serious risks of bias and thus reduce its clinical generalization.

Major Points

1. Performance benchmarking

Your work uses age and gender as additional features added in the final fully connected layer of your BMD estimation model. The decision of giving this additional information remains little explained and a more advanced evaluation procedure would be beneficial for its understanding and adoption in this study. This would allow assessment of potential bias in the dataset and facilitate comparison with other datasets in the literature. As an example, the use of a linear regression with age and gender (similar to how age and gender data is used in your model) would give an estimation of the baseline performance of these variables. In addition, you could blind all image-based features to evaluate the importance of these features to the model's performance. The opposite could also be done to evaluate the model using only image-based features. You could also use feature importance techniques, such as image activation maps (eg. Grad-CAM) and permutation importance (eg. SHAP), to clarify or even explain its decision-making process.

2. Results reporting

Several pieces of information regarding intermediate results have not been validated and/or reported, and may result in a potential bias that could significantly degrade the overall results reported.

A. Experimental setup

You certainly explored various experiments to achieve such results. If this is the case, reporting the best model architecture and configuration would include some survival bias and a significant number of researchers would be highly interested of receiving more information on the range of experiences you have performed, which performed better, which not, and eventually why.

B. Landmark detection performance

You have developed and evaluated the automatic landmark detection model to expert annotations. However, the performance of this model has never been validated, neither in this work nor in any other study. How can you ensure this does not affect the total performance of your automated tool? In addition, for transparency and replication, some information about its performance is necessary otherwise the reported values of your complete system can be questionable.

C. Quality control model configuration for hip radiographs

You used PelviXNet to identify fractured hips. However, you do not mention how you detect

automatically hip implants. As I understand, you did it similarly to vertebral model by using a single model, which is PelviXNet. However, the implant and fracture identification model is rather different from the one reported in your previous work (29). Please clarify and complete. A comment at line 191 is associated to this issue.

D. Quality control model performance for hip radiographs

This model seems to have been trained in similar conditions that it is in ref (29) (comparable datasets with comparable demographic and measuring equipment). Despite the model match these requirements, in ref (29), the model has been validated on a small and rather simple cases (by excluding abnormal cases). In addition, you did not trained your model to identify implants. In this present work, you did not further validated your PelviXNet model. How can you ensure your model would result similarly in your clinical situation where positive labels are different (including implants and with certainly more complex situations of fractures)? Similarly to the landmark detection performance reporting, for transparency and replication, you should report an intermediate validation of this specific task, otherwise your entire automated process can be biased.

E. Quality control model performance for vertebral radiographs

Same question (2-D.) is valid for the vertebral image quality assessment. Despite similarities with your PelviXNet model, you did not report validation performances of this model, which could call into question the whole system based on vertebral images.

F. Extrapolation to other X-ray machines

It would be important to know in which conditions your clinical X-rays were taken, meaning same X-ray machine / Model / brand / acquisition parameters (kVp etc...) Indeed X-ray detectors and acquisition parameters can differ substantially on the market and between practice... would your models be sensitive to variability in those parameters? Did you use data augmentation by creating artifacts to your images?

G. Vertebrae exclusion

In clinical practice, one is supposed to exclude vertebra(e) with more than 1SD difference with adjacent vertebra(e) as it is most likely related to arthrosis etc.. according to ISCD guidelines. This can substantially change the overall BMD and T-score values. How do you take that into account in your models?

3. Metrics used

In the case of imbalanced class cases, such as hip osteoporosis and 10-year of major osteoporotic fractures in your work which present about 20% of positive cases, may result in overestimation of the real model performances. In such context, the use and reporting of precision-recall curve and its associated AUC (AUPRC) would be of high interest. In addition, this would be more aligned with screening problems, in addition to classification metrics such as sensitivity, where positive class is on major importance.

Minor Points

Which Hip BMD did you access and compared with? Total Hip ? Femoral neck ?

Line 84: you are mentioning GE DXA scanner but in lines 319-321 you are describing Hologic scanner. Please correct whatever is true.

Line 144 onward: what are your criteria of "success" here? It is confusing for the reader as not specifically defined.

Line 168: authors should be careful as comparing for example your BMD estimation at the spine/hip from AI on x-ray with QUS is debatable and won't tell you much. Why not comparing it also with similar approaches based on X-ray to estimate BMD studies? Some of those study in a review recently published in JBMR (J Bone Miner Res. 2021 Mar 22. doi: 10.1002/jbmr.4292. Online ahead of print PMID: 33751686 Review. Machine Learning Solutions for Osteoporosis-A Review. By Smets J, Shevroja E, Hügle T, Leslie WD, Hans D.)

Line 178: To avoid any ambiguity, you could precise "for further DXA-BMD diagnosis".

Line 191: Here, the quality control for exclusion of hip replacement implants is not precisely described here.

Line 247: Here I would be more careful and say "should not vary much"

Line 381: In your previous work (41), you trained and evaluated your model on face dataset in an unsupervised manner. In this work, as I understand, you trained your model in a supervised fashion using expert annotations instead of using self-generated landmarks. Some clarification would be necessary here.

Line 483: It is unclear which of the FRAX parameters were used to compute the 10-year risk. All used parameters should be explicitly presented in this section and not only briefly in the limitations

section.

Table 2: You may want to say metrics* instead of matrices.

Figure 3: Please detail explicitly the axis labels to describe the BMD estimated at LS and hip (e.g. for hip calibration plot: y axis= predicted hip BMD, x axis = DXA-measured hip BMD).

REVIEWER COMMENTS

Reviewer #1 (Remarks to the Author):

The study presents a deep-learning based tool to estimate BMD from pelvis radiographs as well as sagittal radiographs of the lumbar spine, which has been packaged together with a simplified version of the FRAX scoring system to predict the risk of osteoporotic fracture. The methods are solid and the deep learning models have been trained on relatively large datasets. The paper is professionally written and presented, and is enjoyable to read.

However, I have a major criticism; the motivation of the study is clear but I do not think that it is valuable enough to grant publication on Nature Communications. The novel method “replaces” DXA for the calculation of the BMD, which is interesting from an academic point of view, and the opportunistic use of radiographs is clinically relevant to some extent; however, I do not see a major clinical advantage in doing that. DXA is cheap, fast, widely available and has a lower X-ray dose in comparison with standard radiographs. I understand the point of the Authors about the fact that “current DXA-based programs screen fewer than one-third of eligible women and one-tenth of eligible men”, but I do not see how this makes “osteoporosis screening based on DXA inadequate”. Obtaining plain radiographs from the general population (other than those needing them for clinical purposes) would be even more difficult, and ethically questionable due to the additional dose of ionizing radiation. DXA remains the golden standard for osteoporosis screening; plain radiographs will not replace DXA in this respect. If a patient needs an investigation of the bone quality, he/she should undergo DXA (or qCT).

Answer:

We thank the reviewer's comment. The study provides proof-of-concept and clinical evidence that AI algorithms can automate BMD prediction using plain radiography for opportunistic screening purposes. The purpose of this study is not to replace, instead, to improve the current DXA-based osteoporosis screening strategy by broadening the screening population.

We answer your concerns point-by-point in the following response:

1. The purpose of opportunistic screening and ethical issues: In contrast to organized/systematic screening for osteoporosis using DXA, our opportunistic screening tools use plain films for other indications. Our tool used plain films that have “already been taken”. A patient may receive a plain film for other clinical reasons, and the physicians viewing the film may not be a specialist or actively surveying for osteoporosis. Thus, our tool may prompt physicians to encourage at-risk individuals to take action. Plain radiographs will not be obtained for osteoporosis screening as a primary indication. Therefore, there should be no ethical concerns here.
2. Supplement to current DXA-based strategy: We agree that the technology described in our paper cannot “replace” DXA. Our automated tool is a supplement to DXA in terms of osteoporosis screening. If patients have access to DXA, they should be screened by DXA as a plain and simple solution. If patients need to understand bone quality and fracture risk, they certainly should seek DXA (or qCT) screening. However, this is not the clinical scenario we are targeting. We aim to use the tool for opportunistic screening for osteoporosis. That is, to provide additional bone quality information to patients who have “already” have a plain film taken for other reasons. We will not encourage patients to use our tool to investigate their bone quality as the “primary” purpose.

Our results show that only a minority of patients having a pelvis (18.6%) or spine plain film (18.2%) have been screened for osteoporosis by DXA. The vast amount of unscreened population exceeds the capacity of DXA in our hospital to process. Our tool provides additional information on the DXA-unscreened population who has plain radiographs of pelvis or lumbar spine. If they have reached the concrete osteoporosis threshold (predicted t score < -2.9) or normal BMD threshold (predicted t score > -2.3), they do not need to take DXA again because our tool has a 95% PPV to classify or 95% NPV to exclude osteoporosis. For the rest of the patients, they can be advised to take DXA subsequently. Our real-world evidence shows that during January-May 2021, 5206 patients with hip or spine radiographs were classified or excluded as osteoporotic with high PPV or NPV for osteoporosis; only 933 patients were advised to take DXA examination. At the same period,

3008 DXA examinations were conducted in the hospital. This is solid and strong evidence indicating the clinical usefulness of our tool to expand the osteoporosis screening population, without additional cost, time and exposure to radiation.

3. The availability of DXA:

The availability of DXA varied widely¹ but in general DXA is of limited availability in developing countries and some European countries. The requirement for assessing and monitoring the treatment of osteoporosis to service practice guidelines has been estimated that in the scenario for screening women with BMD for age ≥ 65 years, 11.21 units of DXA per million population are needed.² However, the joint survey conducted by the International Osteoporosis Foundation (IOF) found a significant regional variation of the DXA scanner density across Asian-Pacific regions, from 1 to 24 DXA machines per million population. A European survey found that 9 EU member states have inadequate provision (less than the 11 DXA units per million population).³ The IOF audit for Latin America found that only Chile and Brazil meet the DXA availability standard and other countries had DXA density ranging from 0.9-6.7 units per million.⁴ In the US, the estimated DXA density was 35.8 units per million population, but DXA utilization was found to be decreasing and at 110 DXA scans per 1000 person-year in post-menopausal women.⁵

From these figures, it is clear that DXA is of limited availability in many areas of the world, especially for developing countries. The availability of plain film is much greater than DXA in these circumstances. Therefore, our tool can be an 'adjunct' tool to provide essential information for patients receiving plain films. In this way, our automated tool to evaluate fracture risk using hip or spine radiographs can help effectively broaden the screening population and increase the number of identifiable high-risk patients.

4. Revision to the manuscript:

We have revised the manuscript to highlight our technology as a tool for opportunistic screening for osteoporosis, as a supplement to the current DXA-based strategy, rather than a primary screening tool in competition with

DXA.

In the first paragraph of the introduction, we emphasize the central role of DXA and FRAX:

“Dual-energy X-ray absorptiometry (DXA) is the currently preferred modality for the measurement of bone mineral density (BMD) in the human hip or lumbar spine, which is an essential component of the fracture risk assessment tool (FRAX) used to estimate the 10-year risk of hip or major osteoporotic fracture.⁶”

In the second paragraph of the introduction, we emphasize the purpose of opportunistic osteoporosis screening:

“Opportunistic screening for osteoporosis using imaging modalities other than DXA is a potential strategy to stratify the unscreened population into distinct risk groups of osteoporosis and fragility fractures. This approach used radiographs ‘already been taken’ for other clinical indications to screen osteoporosis at no additional cost, time, or radiation exposure to patients.”

In the last paragraph of results, we describe the real-world clinical test which is designed to assess the impact of our automatic tool in osteoporosis screening:

“Next, we implemented the tools in the central inference platform connected to the picture archiving and communication system (PACS) in the Chang Gung Memorial Hospital (CGMH, Linkou branch) to study the real impact of our tool to screen osteoporosis. The hospital PACS relayed all newly acquired images to the inference platform daily. In total, 2388 consecutive pelvis (1858 patients, 43.2% women) and 9741 lumbar radiographs (5336 patients, 40.8% women) in those aged 40-90 years were conducted between January and May 2021. The tool excluded 816 pelvis radiographs and 1715 spine radiographs due to poor image quality, inappropriate positions, implants, and fractures that may impede BMD estimation. The percentages of images passing through the entire pipeline and successfully reporting a predicted BMD were 79.0% for pelvis radiographs and 82.3% for spine radiographs. Among these, 5206 (84.8) patients with hip or spine radiographs were classified or excluded as osteoporotic with high PPV or NPV for osteoporosis using thresholds reported in Table 5. Finally, only 933 (15.2%)

patients were advised to take the DXA examination (figure S5). At the same period, 3008 DXA examinations were conducted in CGMH.”

“

Besides, since the models are trained to estimate the BMD based on DXA-measured values, they cannot be “better” than DXA, which has its own limitations. If the models were trained using qCT as ground truth, then I would potentially see an advantage since qCT provides “higher quality” data with respect to both BMD and plain radiographs and it would allow go beyond the state-of-the-art, but this is not the case here.

Answer:

Thanks for the reviewer’s comment. We fully agree that the performance of our tool is a comparison with DXA, so our tool cannot be better than DXA. We did not claim that the performance of our tool surpasses that of DXA; instead, we believe our tool is an adjunct tool to the current osteoporosis screening strategy. As mentioned in the previous response, we should encourage patients to screen for osteoporosis by DXA if possible since it is the current recommended modality to measure BMD. In patients already taken the plain film for other indications, especially in areas with limited access to DXA, the additional information can be valuable information to physicians and patients, enabling them to take action. That is why we design the confidence threshold for our tool and test it in real clinical scenarios. In this way, we can effectively utilize plain films as an adjunct tool to screen osteoporosis, which is fully integrated into the current DXA-based strategy.

We will modify the wording to improve clarity.

Other comments:

- as far as I know, Nature Communications requires peer review of the computer code, but I could not find out how to access the source code. Full access to it should be provided (neural network architecture, training and inference).

Answer:

Thanks for your comments. We have provided a Gigantum project for reviewers to test our model. Data and Code availability will follow what Nature publisher permits. The Nature Communications editor will help the reviewers to access the docker and sample images. The link to the Gigantum repositories:

Docker project: <https://gigantum.com/xraybmd/nc-bmd-cpu>

X-ray dataset: <https://gigantum.com/xraybmd/bmd-data>

QA and BMD models: <https://gigantum.com/xraybmd/bmd-model>

- Abstract: since the main tool discussed in the paper performs a prediction of the BMD value, please report data about the error in its estimation rather than (or in addition to) the accuracy in the identification of osteoporosis/fracture risk. The calculation of the fracture risk is indeed based on the FRAX method, although with modifications and simplifications; its accuracy is therefore driven by the accuracy in estimating the BMD.

Answer:

Thank you. We have revised the abstract accordingly. The reporting of data analysis has been revised to:

“The performance of this tool was evaluated in 5164 and 18175 patients with the pelvis or lumbar spine radiographs and Hologic DXA. The model was well calibrated with minimal bias in the hip (slope = 0.982, calibration-in-the-large = -0.003) and the lumbar spine BMD (slope = 0.978, calibration-in-the-large = 0.003). The area under the precision-recall curve and accuracy were 0.89 and 91.7% for hip osteoporosis, 0.89 and 86.2% for spine osteoporosis, 0.83 and 95.0% for high 10-year major fracture risk, and 0.96 and 90.0% for high hip fracture risk, respectively. Model performance remains robust in datasets based on GE DXA and from the external hospital.”

Reviewer #2 (Remarks to the Author):

This manuscript presents an opportunistic screening tool for osteoporotic fractures in hip and lumbar spine using plain radiographs. The tool has high clinical value since it runs automatically in the background and can identify high risk osteoporotic patients and measure BMD during their routine hospital check-up. It can generate big cost saving for the patients and the hospitals. The tool was trained and tested on a large data set. It was packaged and integrated in a PACS system and was the first to be evaluated in a large trial in real clinical setup. They developed state-of-the-art AI techniques including deep adaptive graph for landmark detection, DenseNet for quality assessment in the tool. The models were robust under different radiograph image quality and achieved high performance. The manuscript is well organized and clearly written.

Answer: Thanks for the reviewer's comment.

Detailed comments:

1) It would be desirable to have more in-depth reviews on computational methods for BMD measurement and fracture detection, including deep learning and traditional image analysis methods.

Answer: We have done a literature review and revised the discussion paragraph 4 as follows:

“Opportunistic osteoporosis screening using other imaging modalities has been reported previously but none had been clinically examined as comprehensive as our study. The best-studied strategy is the use of abdominal CT to predict BMD,^{7, 8, 9} classify osteoporosis based on CT attenuation,¹⁰ simulated BMD,^{8, 9} T-score,⁷ or detection of osteoporotic fractures;¹¹ or use imaging biomarkers to predict the risk of fractures.¹² Julien Smets et al. reviewed machine learning solutions for osteoporosis.¹³ Among five studies using CT scans to predict BMD, the best correlation coefficient between estimated and CT-simulated spine BMD was 0.94.¹⁴

An earlier study compared the CT Hounsfield units over a manually annotated ROI involving vertebral body trabecular bone with its paired DXA T-score; this approach for detection of osteoporosis yielded an AUC of 0.83.¹⁰ Deep learning-based models provided a better correlation between predicted and reference values, but were only validated in small datasets.^{7,8,9} A larger study testing the performance of simulated T-scores on a larger dataset of 1843 CT-DXA pairs achieved an accuracy of 82% to detect osteoporosis.⁷ This algorithm was integrated with VCF identification and CT trabecular density as biomarkers, and its performance for the prediction of 5-year fracture risks was compared favorably with the performance of FRAX-NB.¹² Osteoporosis and fragility fracture risk have also been assessed on dental,^{15,16} hip,^{17,18} and spine radiographs,^{16,19} and magnetic resonance imaging.²⁰ However, only three were validated against standard DXA-based hip or spine BMD. The best AUROC was 0.92 for hip¹⁷ and 0.73 for spine osteoporosis classification using small testing sets (131 and 345 patients, respectively).¹⁹ These studies demonstrated the feasibility of using non-DXA modalities to screen osteoporosis, although the applicability and usability of such tools in real clinical settings are questionable.”

2) *Why exclude data obtained using GE DXA scanner?*

Answer: Thanks for the reviewer’s comment.

1. Different BMD numeric results on machines by different manufacturers: The Hologic and GE provide different BMD values for the same patient. As the following chart shows, for the same reference t score (NHANES III), GE and Hologic DXA report BMD values with a systemic bias. In our cohort, some patient was scanned by Hologic and some by the GE DXA. Therefore, if we mixed the data, the performance was dampened in our previous experiments.

2. Choosing the Hologic system as our primary training and testing data:

In Chang Gung Memorial Hospital, the GE Lunar iDXA system is relatively new to the hospital. The paired DXA-radiograph data are much more abundant for the Hologic system. Therefore, we trained the model based on the Hologic measurement.

3. Conversion of GE BMD values to Hologic values:

According to the white paper “Practical Considerations When Replacing a DXA system” provided by Hologic Inc., GE BMD values can be converted to the Hologic values by the following equations: We converted the GE values to Hologic and compared the performance of algorithm-predicted BMD.

Lumbar Spine: Hologic BMD= 0.918 x Lunar BMD – 0.038

Total Hip: Hologic BMD= 0.971 x Lunar BMD – 0.037

4. Testing on GE BMD results:

We identified 2060 patients with paired GE DXA-pelvis radiographs and 3692 patients with paired GE DXA-lumbar spine radiographs to validate our tool. The performance is also robust but slightly lower than the Hologic data. A standalone

training based on GE data may improve this. We add the GE performance data to the results:

“We identified 2060 patients with paired GE DXA-pelvis radiographs and 3692 patients with paired GE DXA-lumbar spine radiographs (table S2). The GE BMD values were converted to Hologic values using the manufacturer-provided equations (table S3). Supplementary Table S4 summarizes the model performance by comparing model-predicted BMD and GE DXA-measured BMD and table S5 summarized the discriminatory performance. The Pearson’s correlation coefficients between GE DXA-measured and model-predicted BMD were 0.90 for the hip and 0.89 for the hip and lumbar spine. The model remains robust with good linear correlation, calibration and minimal bias across different age and sex strata. The discriminatory performance is also excellent, with an AUPRC of 0.87 for the hip and 0.89 for the spine model. We further test our tool using 34 pairs of GE DXA-hip radiographs and 179 pairs of DXA-lumbar spine radiographs from the Wuhan Hospital of Traditional Chinese Medicine. The Pearson correlation coefficient was 0.93 for the hip model and 0.86 for the spine model. “

3) How do you split the training and test sets?

Answer: As figure 1 shows, we split the data randomly after a series of inclusion and exclusion criteria. These patients were randomly split into the testing and training set by Simple Random Sampling. In simple random sampling, each unit has an equal probability of selection, and sampling is without replacement. Without-replacement sampling means that a unit cannot be selected more than once. We revised the data flow chart (figure S1) to improve the clarity of the data preparation process.

4) Line 98, please specify what unsuitable vertebrae are. Since 30% of vertebrae are excluded, is the quality assessment automatic? What will happen to those excluded vertebrae? A warning will be issued?

Answer:

The automated quality assessment procedure for spine radiographs is performed in three steps: implant and VCF detection, six-point morphology analysis and assessment for T-score of nearby vertebrae. Yes, the quality assessment is fully automated. These excluded vertebrae were not used to predict BMD and t score. In our UI, the excluded vertebrae are highlighted in red as a visualization and warning. The sample visualization as follows:

5) Line 99, how were BMD measured? By how many operators? Is there inter-operator variability?

Answer:

The densitometer is operated by multiple radiology technicians per manufacturers' instructions. In CGMH, all technicians need to be certified to operate the densitometer. The scans were analyzed following recommendations issued by the Taiwan Radiological Society²¹ (amended from International Society for Clinical Densitometry, ISCD).²² The recommendation in brief: 1. Completion of ISCD training

course and at least 120 cases of DXA examination before independent operation of DXA. 2. The standard phantom measurement needs to perform regularly, and the measured values need to fall within 1.5% error. 3. All DXA technicians need to establish a personal least significant change (LSC). The current recommendation is that the personal LSC should be within 5.3% for the lumbar spine, 5.0% for the whole hip, and 6.9% for hip neck BMD measurement.

We have put the information into the main methods and in the supplementary material as follows:

In the main methods:

“The scans were analyzed following recommendations issued by the Taiwan Radiological Society,²¹ which is amended from the International Society for Clinical Densitometry, ISCD, described in the supplementary material (supplementary method).²²”

In the supplementary method:

“The densitometer is operated by multiple radiology technicians. In CGMH, all technicians need to be certified to operate the densitometer. The scans were analyzed following recommendations issued by the Taiwan Radiological Society²¹ (amended from International Society for Clinical Densitometry, ISCD).²² The recommendation briefly: 1. All DXA technicians need to complete the ISCD training course (basic and advanced) and operate at least 120 cases of DXA examination before the independent operation of DXA. 2. All technicians need to obtain the SOP provided by the manufacturer and conduct BMD measurements accordingly. 3. All DXA scanners need to have a detailed SOP at the examination site, which needs to be updated regularly and reviewed by relevant professionals. 4. All DXA scanners need to comply with the local radiation safety guideline. 5. Spine phantom BMD measurement is performed regularly to document the stability of DXA performance over time. BMD values must be maintained within an error of $\pm 1.5\%$. A monitoring plan is needed for a correction approach when the error has been exceeded. 6. All DXA technicians need to establish a personal least significant change (LSC). The

current recommendation is that the personal LSC should be within 5.3% for the lumbar spine, 5.0% for the whole hip, and 6.9% for hip neck BMD measurement.”

,

6) Line 151, among the 90.4% of radiograph with predicted BMD, do you assess the accuracy of BMD value? Do you calculate the distribution of BMD? Is it in line with those in the training set?

Answer:

In this experiment, we installed the docker in the central inference server to see how many of these images can predict BMD using our algorithm. We do not have data of linked BMD to ascertain the accuracy. This real-world test aims to assess the impact of the model to classify osteoporosis. We have revised this part:

“Next, we implemented the tools in the central inference platform connected to the picture archiving and communication system (PACS) in the Chang Gung Memorial Hospital (CGMH, Linkou branch) to study the real impact of our tool to screen osteoporosis. The hospital PACS relayed all newly acquired images to the inference platform daily. In total, 2388 consecutive pelvis (1858 patients, 43.2% women) and 9741 lumbar radiographs (5336 patients, 40.8% women) in those aged 40-90 years were conducted between January and May 2021. The tool excluded 816 pelvis radiographs and 1715 spine radiographs due to poor image quality, inappropriate positions, implants, and fractures that may impede BMD estimation. The percentages of images passing through the entire pipeline and successfully reporting a predicted BMD were 79.0% for pelvis radiographs and 82.3% for spine radiographs. Among these, 5206 (84.8) patients with hip or spine radiographs were classified or excluded as osteoporotic with high PPV or NPV for osteoporosis using thresholds reported in Table 5. Finally, only 933 (15.2%) patients were advised to take the DXA examination (figure S5). At the same period, 3008 DXA examinations were conducted in CGMH.”

7) Are all the radiographs acquired from the same machine? If not, any performance difference?

Answer:

The CGMH has seven hospitals, and most of the pelvis and lumbar spine radiographs are produced by Cannon CDXI 710C and Shimadzu MUX-100H (87.14% of spine radiographs). The most common kVp for the lumbar spine is between 70-95 (around 80%). Technicians will adjust Kvp to ensure good tissue penetration and image quality. We prepared a summary statistics table to show the hip and lumbar spine kVp and the correlation between DXA-measured and model-predicted BMD. In general, the performance did not change. The following table compare the model performance in different kVp setting or radiograph machines.

Hip radiographs				Spine radiographs			
kVp distribution	n	%	Correlation coefficient	kVp distribution	n	%	Correlation coefficient
60-69 kV	1245	24.11	0.909	70-80 kV	5157	28.37	0.896
70-74 kV	995	19.27	0.921	90 kV	4637	25.51	0.889
75 kV	1394	26.99	0.917	95 kV	4747	26.12	0.898
Other	1530	29.63	0.922	Other	3634	19.99	0.889
Machine type	n	%	Correlation coefficient	Machine type	n	%	Correlation coefficient
Canon CDXI 710C	2576	49.88	0.919	Canon CDXI 710C	12337	67.88	0.896
Shimadzu MUX-100H	1161	22.48	0.914	Shimadzu MUX-100H	3501	19.26	0.885
Other	1427	27.63	0.917	Other	2337	12.86	0.887

8) Line 381, is the data used to train DAG models part of the training set listed in Table 1? If not, please specify.

Answer:

The data used to train DAG models are separate from the training/testing data listed in Table 1. As our data flow chart shows, the training and testing data for DAG (9111 patients with hip radiographs and 2641 patients with spine radiographs) are extracted from the original plain dataset, which were not included in the dataflow for BMD estimation models. We amended the data flow chart (figure S1) to highlight the separate dataset for QA model development.

9) Line 444, does the DXA-based BMD measurement consider age and sex?

Answer:

The DXA-based BMD measure does not consider age and sex. Our experiment shows that the addition of age and sex did not improve the model performance. The results of these experiments are shown in Table S7-8.

“We evaluated multiple backbone networks (i.e., VGG-11, VGG-16, ResNet-18, ResNet-34) in earlier experiments and empirically found that VGG-16 and ResNet-34 produce the best BMD prediction results for spine and hip BMD prediction, respectively. The model using only image-based features already performs strongly, and the addition of age and gender does not improve the model’s performance. Therefore, we choose to use the VGG16 backbone without age and sex in later model development (Table S7-S8).”

10) Line 450, how to label vertebra index? How accurate?

Answer:

The DAG landmark detector detects landmarks of L1-L4 vertebrae with their vertebra indexes. The spine landmark DAG model is validated on 290 spine X-ray images and

reports 1.22+-3.23mm localization error. The hip landmark DAG model is validated on 876 pelvic X-ray images and reports 4.29+/-3.29mm localization error. We add these results to the revision.

11) Since all data are from a single hospital center. How good would this model be generalized for data from a new center? Do you need retraining or transfer learning? Please discuss.

Answer:

Chang Gung Memorial Hospital has 7 hospitals located across Taiwan. We believe our model can be generalized. We tested our model using data from the external source (Wuhan Hospital of Traditional Chinese Medicine) and found an excellent correlation coefficient of 0.93 for the hip and 0.86 for the spine model. However, we believe a small set of local data would optimize the process and improve the accuracy.

CCMH serves patients with a higher severity; as in our limitation discussed, the model has exposed to more complicated cases than ordinary patients who require osteoporosis screening. Therefore, our model can adapt well to the most clinical scenario.

Since our model is based on plain radiographs, the quality of such films is essential. The current model uses all digital images. Scanned images may dampen the performance.

Reviewer #3 (Remarks to the Author):

This work describes a fully automated system using age and sex information in addition to radiographic images of the hip or spine to estimate BMD and infer risk via osteoporosis and FRAX measurements. The document is well written and is on average clearly detailed. Achieved performances were highly promising and would be of great impact in the high risk population screening. However, several points

concerning performance benchmarking, results reporting and metrics used remains to be addressed. Overall, the results should be more mitigated by the serious limitations observed, particularly regarding the lack of intermediate validation, which leaves room for serious risks of bias and thus reduce its clinical generalization.

Major Points

1. Performance benchmarking

Your work uses age and gender as additional features added in the final fully connected layer of your BMD estimation model. The decision of giving this additional information remains little explained and a more advanced evaluation procedure would be beneficial for its understanding and adoption in this study. This would allow assessment of potential bias in the dataset and facilitate comparison with other datasets in the literature. As an example, the use of a linear regression with age and gender (similar to how age and gender data is used in your model) would give an estimation of the baseline performance of these variables. In addition, you could blind all image-based features to evaluate the importance of these features to the model's performance. The opposite could also be done to evaluate the model using only image-based features. You could also use feature importance techniques, such as image activation maps (eg. Grad-CAM) and permutation importance (eg. SHAP), to clarify or even explain its decision-making process.

Answer:

In the revision, we significantly expanded the sizes of the datasets. Using the expanded datasets, we found that the model using only image-based features already performs strongly, and the use of age and gender does not further improve the model's performance. Therefore, we did not add the age and sex into the final model. We revised the method section as follows:

"We evaluated multiple backbone networks (i.e., VGG-11, VGG-16, ResNet-18, ResNet-34) in earlier experiments and empirically found that VGG-16 and ResNet-34 produce the best BMD prediction results for spine and hip BMD prediction, respectively. The model using only image-based features already performs strongly, and the addition of age and gender does not improve the model's performance. Therefore, we choose to use the VGG16 backbone without the addition of age and sex (Table S6-S7)."

2. Results reporting

Several pieces of information regarding intermediate results have not been validated and/or reported, and may result in a potential bias that could significantly degrade the overall results reported.

A. Experimental setup

You certainly explored various experiments to achieve such results. If this is the case, reporting the best model architecture and configuration would include some survival bias and a significant number of researchers would be highly interested of receiving more information on the range of experiences you have performed, which performed better, which not, and eventually why.

Answer: In the revision, we benchmarked several models (VGG11, VGG16, ResNet18, ResNet34) for BMD estimation. We performed 4-fold cross validation for each backbone model and selected the best model based on the validation results for each fold. Then the selected models are ensembled for evaluation on the test set. We found that different backbone models achieve different performances. The benchmark results are added to the supplementary material (S6-S7). We revised the main method to describe the benchmarking:

“We evaluated multiple backbone networks (i.e., VGG-11, VGG-16, ResNet-18, ResNet-34) in earlier experiments and empirically found that VGG-16 and ResNet-34 produce the best BMD prediction results for spine and hip BMD prediction, respectively. The model using only image-based features already performs strongly, and the addition of age and gender does not improve the model’s performance. Therefore, we choose to use the VGG16 backbone without the addition of age and sex (Table S6-S7).”

B. Landmark detection performance

You have developed and evaluated the automatic landmark detection model to expert annotations. However, the performance of this model has never been validated, neither in this work nor in any other study. How can you ensure this does not affect the total performance of your automated tool? In addition, for transparency and replication, some information about its performance is necessary

otherwise the reported values of your complete system can be questionable.

Answer:

The spine landmark DAG model is validated on 290 spine X-ray images and reports 1.22+/-3.23mm localization error. The hip landmark DAG model is validated on 876 pelvic X-ray images and reports 4.29+/-3.29mm localization error. We add these results to the revision.

“The radiographs used to train the DAG models are excluded from the test sets used to evaluate the BMD estimation models. The DAG models are evaluated on 876 and 290 pelvic and spine radiographs and report 4.29+/-3.29mm and 1.22+/-3.23mm localization errors, respectively.”

C. Quality control model configuration for hip radiographs

You used PelviXNet to identify fractured hips. However, you do not mention how you detect hip implants automatically. As I understand, you did it similarly to the vertebral model by using a single model, which is PelviXNet. However, the implant and fracture identification model is rather different from the one reported in your previous work (29). Please clarify and complete. A comment at line 191 is associated to this issue.

Answer:

We trained an implant detection model using the same network architecture as the PelviXNet. A description of the hip implant detection model is added to the Method section. The spine implant/VCF detection model has the same network architecture as the PelviXNet and is trained using the same mechanism. The original PelviXNet was trained using supervision masks generated from point annotations. The implant/VCF detection model is also trained using supervision masks generated using vertebra polygons. We have added the information into the methods as follows:

“To evaluate the performance of the spine radiograph QA module, we randomly selected 200 spine radiographs from the test set and manually labeled implant and VCF. The implant and VCF detection module report 91.5% and 93.2% sensitivity and

99.5% and 91.5% specificity. Some mild VCFs are not detected by the VCF detection module alone.”

D. Quality control model performance for hip radiographs

This model seems to have been trained in similar conditions that it is in ref (29) (comparable datasets with comparable demographic and measuring equipment). Despite the model match these requirements, in ref (29), the model has been validated on a small and rather simple cases (by excluding abnormal cases). In addition, you did not trained your model to identify implants. In this present work, you did not further validated your PelviXNet model. How can you ensure your model would result similarly in your clinical situation where positive labels are different (including implants and with certainly more complex situations of fractures)? Similarly to the landmark detection performance reporting, for transparency and replication, you should report an intermediate validation of this specific task, otherwise your entire automated process can be biased.

Answer:

In ref (29), the PelviXNet model has been validated on all PXR images taken in the trauma center of CGMH in 2017 without excluding any case. Therefore, the testing scenario represents the population of trauma patients, which contains complex and challenging cases (including many implants, as shown in Fig. 3 in ref 29). We note that PelviXnet is used to detect fractures only, and another model is trained to detected implants. We have revised the manuscript to describe the training setup clearly.

“In addition to detecting hip fracture, we trained another network identical to that of PelviXNet using 2973 pelvic radiographs to detect implants. The maximum responses of the fracture and implant detection networks in the hip ROI are calculated as the classification scores for hip fracture and implant, respectively. The fracture detection model, PelviXNet, was evaluated on 1888 pelvic radiographs covering various medical conditions (e.g., implants and periprosthetic fracture) and reports 92.4% sensitivity and 90.8% specificity. The implant detection model was evaluated on 719 randomly selected pelvic radiographs and reports 99.9% sensitivity and 99.7% specificity.”

E. Quality control model performance for vertebral radiographs

Same question (2-D.) is valid for the vertebral image quality assessment. Despite similarities with your PelviXNet model, you did not report validation performances of this model, which could call into question the whole system based on vertebral images.

Answers:

We added validation performances of the spine radiograph QA module in the revision:

“We randomly selected 200 spine radiographs from the test set and manually labeled implant and VCF. The implant and VCF detection module report 91.5% and 93.2% sensitivity and 99.5% and 91.5% specificity, respectively.”

F. Extrapolation to other X-ray machines

*It would be important to know in which conditions your clinical X-rays were taken, meaning same X-ray machine / Model / brand / acquisition parameters (kVp etc...)
Indeed X-ray detectors and acquisition parameters can differ substantially on the market and between practice... would your models be sensitive to variability in those parameters?*

Answers:

The CGMH has 7 hospitals, and most of the pelvis and lumbar spine radiographs are produced by Cannon CDXI 710C and Shimadzu MUX-100H (87.14% of spine radiographs). The most common kVp for the lumbar spine is between 70-95 (around 80%). Technicians will adjust Kvp to ensure good tissue penetration and image quality. We prepared a summary statistics table to show the hip and lumbar spine kVp and the correlation between DXA-measured and model-predicted BMD. In general, the performance did not change.

Hip radiographs				Spine radiographs			
kVp distribution	n	%	Correlation coefficient	kVp distribution	n	%	Correlation coefficient
60-69 kV	1245	24.11	0.909	70-80 kV	5157	28.37	0.896
70-74 kV	995	19.27	0.921	90 kV	4637	25.51	0.889
75 kV	1394	26.99	0.917	95 kV	4747	26.12	0.898
Other	1530	29.63	0.922	Other	3634	19.99	0.889
Machine type	n	%	Correlation coefficient	Machine type	n	%	Correlation coefficient
Canon CDXI 710C	2576	49.88	0.919	Canon CDXI 710C	12337	67.88	0.896
Shimadzu MUX-100H	1161	22.48	0.914	Shimadzu MUX-100H	3501	19.26	0.885
Other	1427	27.63	0.917	Other	2337	12.86	0.887

Since our training and testing data are from diverse clinical images and the sample size is large, we believe our model can cope with these clinical variations. In addition, we conducted the external validation using data from Wuhan hospital and the model still performs well.

Did you use data augmentation by creating artifacts to your images?

We performed data augmentation during training to cope with the possible variations in the X-ray imaging conditions. The data augmentation includes: 1) color jittering on both the brightness (+/-0.2) and contrast (+/-0.2). 2) random up-down and left-right flipping, 3) random affine transformation (rotation +/-30, shear +/- 0.2, translation +/- 25 pixels, scaling +/-10%).

G. Vertebrae exclusion

In clinical practice, one is supposed to exclude vertebra(e) with more than 1SD difference with adjacent vertebra(e) as it is most likely related to arthrosis etc.. according to ISCD guidelines. This can substantially change the overall BMD and T-score values. How do you take that into account in your models?

Answer:

We thank the reviewer's comment. We removed the data with a one sd difference with the neighboring vertebrae, and only those with at least 2 analyzable vertebrae were included in the analysis.

“Quality assessment of spine radiographs: The adult official positions of the International Society for Clinical Densitometry (ISCD) advise to exclude vertebrae that is clearly abnormal and non-assessable or has a more than a 1.0 T-score difference between the vertebra in question and adjacent vertebrae.²³ Therefore the automated quality assessment procedure for spine radiographs in three steps: implant and VCF detection, six-point morphology analysis and assessment for T-score of nearby vertebrae.”

3. Metrics used

In the case of imbalanced class cases, such as hip osteoporosis and 10-year of major osteoporotic fractures in your work which present about 20% of positive cases, may result in overestimation of the real model performances. In such context, the use and reporting of precision-recall curve and its associated AUC (AUPRC) would be of high interest. In addition, this would be more aligned with screening problems, in addition to classification metrics such as sensitivity, where positive class is on major importance.

Answer:

Thanks for the reviewer's comment. We have reported AUPRC throughout the classification tasks.

Minor Points

Which Hip BMD did you access and compared with? Total Hip ? Femoral neck ?

Answer: Total hip BMD.

Line 84: you are mentioning GE DXA scanner but in lines 319-321 you are describing Hologic scanner. Please correct whatever is true.

Answer: We used the GE data for separate testing. The results of GE data were described as follows:

“We identified 2060 patients with paired GE DXA-pelvis radiographs and 3692 patients with paired GE DXA-lumbar spine radiographs (table S2). The GE BMD values were converted to Hologic values using the manufacturer-provided equations (table S3). Supplementary Table S4 summarizes the model performance by comparing model-predicted BMD and GE DXA-measured BMD and table S5 summarized the discriminatory performance. The Pearson’s correlation coefficients between GE DXA-measured and model-predicted BMD were 0.90 for the hip and 0.89 for the hip and lumbar spine. The model remains robust with good linear correlation, calibration and minimal bias across different age and sex strata. The discriminatory performance is also excellent, with an AUPRC of 0.87 for the hip and 0.89 for the spine model. We further test our tool using 34 pairs of GE DXA-hip radiographs and 179 pairs of DXA-lumbar spine radiographs from the Wuhan Hospital of Traditional Chinese Medicine. The Pearson correlation coefficient was 0.93 for the hip model and 0.86 for the spine model. ”

Line 144 onward: what are your criteria of “success” here? It is confusing for the reader as not specifically defined.

Answer:

Success means the images pass through the entire pipeline and successfully report a predicted BMD. We conducted this real-world experiment to (1) show our pipeline can be successfully linked to the PACS system and function reasonably and (2) To

estimate how many people with plain radiographs can be benefited from BMD prediction using our model. However, this experiment is limited to the plain radiographs; we have no access to the linked DXA data. In addition to the success rate, we also examined the percentage of these patients being classified into different categories. We have explicitly defined the meaning of success but avoid the wording of success rate. We also created figure S5 to improve clarity:

“Next, we implemented the tools in the central inference platform connected to the picture archiving and communication system (PACS) in the Chang Gung Memorial Hospital (CGMH, Linkou branch) to study the real impact of our tool to screen osteoporosis. The hospital PACS relayed all newly acquired images to the inference platform daily. In total, 2388 consecutive pelvis (1858 patients, 43.2% women) and 9741 lumbar radiographs (5336 patients, 40.8% women) in those aged 40-90 years were conducted between January and May 2021. The tool excluded 816 pelvis radiographs and 1715 spine radiographs due to poor image quality, inappropriate positions, implants, and fractures that may impede BMD estimation. The percentages of images passing through the entire pipeline and successfully reporting a predicted BMD were 79.0% for pelvis radiographs and 82.3% for spine radiographs. Among these, 5206 (84.8) patients with hip or spine radiographs were classified or excluded as osteoporotic with high PPV or NPV for osteoporosis using thresholds reported in Table 5. Finally, only 933 (15.2%) patients were advised to take DXA examination (figure S5). At the same period, 3008 DXA examinations were conducted in CGMH.”

Line 168: authors should be careful as comparing for example your BMD estimation at the spine/hip from AI on x-ray with QUS is debatable and won't tell you much. Why not comparing it also with similar approaches based on X-ray to estimate BMD studies? Some of those study in a review recently published in JBMR (J Bone Miner Res. 2021 Mar 22. doi: 10.1002/jbmr.4292. Online ahead of print PMID: 33751686 Review. Machine Learning Solutions for Osteoporosis-A Review. By Smets J, Shevroja E, Hügle T, Leslie WD, Hans D.)

Answer:

Thanks for the reviewer's advice. We have removed the description of the comparison with QUS. We have summarized the relevant information in this part of the discussion as follows:

“Opportunistic osteoporosis screening using other imaging modalities has been reported previously but none had been clinically examined as comprehensive as our study. The best-studied strategy is the use of abdominal CT to predict BMD,^{7,8,9} classify osteoporosis based on CT attenuation,¹⁰ simulated BMD,^{8,9} T-score,⁷ or detection of osteoporotic fractures;¹¹ or use imaging biomarkers to predict the risk of fractures.¹² Julien Smets et al. reviewed machine learning solutions for osteoporosis.¹³ Among five studies using CT scans to predict BMD, the best correlation coefficient between estimated and CT-simulated spine BMD was 0.94.¹⁴ An earlier study compared the CT Hounsfield units over a manually annotated ROI involving vertebral body trabecular bone with its paired DXA T-score; this approach for detection of osteoporosis yielded an AUC of 0.83.¹⁰ Deep learning-based models provided a better correlation between predicted and reference values, but were only validated in small datasets.^{7,8,9} A larger study testing the performance of simulated T-scores on a larger dataset of 1843 CT-DXA pairs achieved an accuracy of 82% to detect osteoporosis.⁷ This algorithm was integrated with VCF identification and CT trabecular density as biomarkers, and its performance for the prediction of 5-year fracture risks was compared favorably with the performance of FRAX-NB.¹² Osteoporosis and fragility fracture risk have also been assessed on dental,^{15,16} hip,^{17,18} and spine radiographs,^{16,19} and magnetic resonance imaging.²⁰ However, only three were validated against standard DXA-based hip or spine BMD. The best AUROC

was 0.92 for hip¹⁷ and 0.73 for spine osteoporosis classification using small testing sets (131 and 345 patients, respectively).¹⁹ These studies demonstrated the feasibility of using non-DXA modalities to screen osteoporosis, although the applicability and usability of such tools in real clinical settings are questionable.”

Line 178: To avoid any ambiguity, you could precise "for further DXA-BMD diagnosis".

Answer: We have revised the sentence accordingly.

Line 191: Here, the quality control for exclusion of hip replacement implants is not precisely described here.

Answer:

We have added a description of the hip implant quality control module:

“In addition to detecting hip fracture, we trained another network identical to that of PelviXNet using 2973 pelvic radiographs to detect implants. The maximum responses of the fracture and implant detection networks in the hip ROI are calculated as the classification scores for hip fracture and implant, respectively. The fracture detection model, PelviXNet, was evaluated on 1888 pelvic radiographs covering various medical conditions (e.g., implants and periprosthetic fracture) and reports 92.4% sensitivity and 90.8% specificity. The implant detection model was evaluated on 719 randomly selected pelvic radiographs and reports 99.9% sensitivity and 99.7% specificity.”

Line 247: Here I would be more careful and say "should not vary much"

Answer: Thanks for reviewer’s advice. We have revised it accordingly/.

Line 381: In your previous work (41), you trained and evaluated your model on face dataset in an unsupervised manner. In this work, as I understand, you trained your model in a supervised fashion using expert annotations instead of using

self-generated landmarks. Some clarification would be necessary here.

Answer:

We apologize that there was an error in the reference. Ref (41) should be replaced with Li, Weijian, et al. "Structured landmark detection via topology-adapting deep graph learning." European Conference on Computer Vision (9): 266-283, arXiv preprint arXiv:2004.08190 (2020). This paper describes the DAG method, where landmark detection is trained in supervised learning. The citation number in the revised manuscript is 35.

Line 483: It is unclear which of the FRAX parameters were used to compute the 10-year risk. All used parameters should be explicitly presented in this section and not only briefly in the limitations section.

Answer:

Thanks for reviewer's comment. We have updated the FRAX parameters used in the model in results.

"The median FRAX 10-year major fracture (8.84% vs. 8.76%, $p=0.24$) and hip fracture risks (2.48% vs. 2.46%, $p = 0.06$) did not significantly differ when scores were based on the predicted BMD (FRAX-PB) or measured BMD (FRAX-MB) plus clinical parameters (age, sex, height and weight)."

In the method:

"We also compared the risks of 10-year hip and major osteoporotic fractures (<https://www.sheffield.ac.uk/FRAX/>) using only clinical parameters (FRAX-NB, age, sex, weight and height), clinical parameters and DXA-measured BMD (FRAX-MB) or predicted BMD (FRAX-PB). We also conducted a real-world test on consecutive patients to prove the clinical applicability of our tool and its impact on osteoporosis screening strategy."

Table 2: You may want to say metrics instead of matrices.*

Answer: Thanks for your advice, we have revised the typos throughout the manuscript accordingly.

Figure 3: Please detail explicitly the axis labels to describe the BMD estimated at LS and hip (e.g. for hip calibration plot: y axis= predicted hip BMD, x axis = DXA-measured hip BMD).

Answer: Thanks for your advice. We have changed the x- and y-axis titles accordingly.

Reference

1. Clynes MA, *et al.* Bone densitometry worldwide: a global survey by the ISCD and IOF. *Osteoporos Int* **31**, 1779-1786 (2020).
2. Kanis JA, Johnell O. Requirements for DXA for the management of osteoporosis in Europe. *Osteoporos Int* **16**, 229-238 (2005).
3. Hernlund E, *et al.* Osteoporosis in the European Union: medical management, epidemiology and economic burden. A report prepared in collaboration with the International Osteoporosis Foundation (IOF) and the European Federation of Pharmaceutical Industry Associations (EFPIA). *Arch Osteoporos* **8**, 136 (2013).
4. Harvey NC, *et al.* Mind the (treatment) gap: a global perspective on current and future strategies for prevention of fragility fractures. *Osteoporos Int* **28**, 1507-1529 (2017).
5. Overman RA, Farley JF, Curtis JR, Zhang J, Gourlay ML, Deal CL. DXA Utilization Between 2006 and 2012 in Commercially Insured Younger Postmenopausal Women. *J Clin Densitom* **18**, 145-149 (2015).
6. Kanis JA, McCloskey EV, Johansson H, Oden A, Strom O, Borgstrom F. Development and use of FRAX in osteoporosis. *Osteoporos Int* **21 Suppl 2**, S407-413 (2010).
7. Krishnaraj A, *et al.* Simulating Dual-Energy X-Ray Absorptiometry in CT Using Deep-Learning Segmentation Cascade. *J Am Coll Radiol* **16**, 1473-1479 (2019).
8. Yasaka K, Akai H, Kunimatsu A, Kiryu S, Abe O. Prediction of bone mineral density from computed tomography: application of deep learning with a convolutional neural network. *Eur Radiol* **30**, 3549-3557 (2020).
9. Fang Y, *et al.* Opportunistic osteoporosis screening in multi-detector CT images using deep convolutional neural networks. *Eur Radiol*, (2020).
10. Pickhardt PJ, Pooler BD, Lauder T, del Rio AM, Bruce RJ, Binkley N. Opportunistic screening for osteoporosis using abdominal computed tomography scans obtained for other indications. *Ann Intern Med* **158**,

588-595 (2013).

11. Tomita N, Cheung YY, Hassanpour S. Deep neural networks for automatic detection of osteoporotic vertebral fractures on CT scans. *Comput Biol Med* **98**, 8-15 (2018).
12. Dagan N, *et al.* Automated opportunistic osteoporotic fracture risk assessment using computed tomography scans to aid in FRAX underutilization. *Nat Med* **26**, 77-82 (2020).
13. Smets J, Shevroja E, Hugel T, Leslie WD, Hans D. Machine Learning Solutions for Osteoporosis-A Review. *J Bone Miner Res* **36**, 833-851 (2021).
14. Gonzalez G, Washko GR, Estepar RSJ. Deep learning for biomarker regression: application to osteoporosis and emphysema on chest CT scans. *Proc SPIE Int Soc Opt Eng* **10574**, (2018).
15. Kavitha MS, Asano A, Taguchi A, Kurita T, Sanada M. Diagnosis of osteoporosis from dental panoramic radiographs using the support vector machine method in a computer-aided system. *BMC Med Imaging* **12**, 1 (2012).
16. Lee KS, Jung SK, Ryu JJ, Shin SW, Choi J. Evaluation of Transfer Learning with Deep Convolutional Neural Networks for Screening Osteoporosis in Dental Panoramic Radiographs. *J Clin Med* **9**, (2020).
17. Yamamoto N, *et al.* Deep Learning for Osteoporosis Classification Using Hip Radiographs and Patient Clinical Covariates. *Biomolecules* **10**, (2020).
18. Sathagirivasan V, Anburajan M. Diagnosis of osteoporosis by extraction of trabecular features from hip radiographs using support vector machine: an investigation panorama with DXA. *Comput Biol Med* **43**, 1910-1919 (2013).
19. Zhang B, *et al.* Deep learning of lumbar spine X-ray for osteopenia and osteoporosis screening: A multicenter retrospective cohort study. *Bone* **140**, 115561 (2020).
20. Ferizi U, *et al.* Artificial Intelligence Applied to Osteoporosis: A Performance Comparison of Machine Learning Algorithms in Predicting Fragility Fractures

From MRI Data. *J Magn Reson Imaging* **49**, 1029-1038 (2019).

21. Society TR. Best Practices for Dual-Energy X-ray Absorptiometry.) (2017).
22. Lewiecki EM, *et al.* Best Practices for Dual-Energy X-ray Absorptiometry Measurement and Reporting: International Society for Clinical Densitometry Guidance. *J Clin Densitom* **19**, 127-140 (2016).
23. Densitometry. TISfC. The adult official positions of the ISCD as updated in 2019.) (2019).

Reviewers' Comments:

Reviewer #1:

Remarks to the Author:

The Authors submitted a remarkably extensive and detailed revised version which addresses all my comments in a satisfactory way. Although in general I agree with the Authors about the comments about the value of opportunistic screening, I am still convinced that this work has a significant but relatively limited clinical relevance and therefore I think that it would better fit a specialty journal about orthopaedics rather than Nature Communications. However, it is technically very sound, professionally presented and relevant to a wide audience and I have therefore no objections against its publication in this journal.

Reviewer #2:

Remarks to the Author:

The revised paper has addressed almost all my concerns. Several issues have been clarified and a couple new experiments have been done to further validate the method. A more thorough literature review has been given. A new experiment was done on GE DXA data. A performance comparison was done on different radiograph machines.

Reviewer #3:

Remarks to the Author:

The authors have addressed my comments appropriately.

I have also checked on the code/dataset but found some in-accuracy in the outcome regarding landmarks for both spine and hip.

I will mention them to the authors underneath as it can be address or at least commented in the papers.

Frankly speaking, the work is nice and well done but I do not know what is your "acceptance threshold" for your journal.

I would be tempted to accept as it is with the minor comments addressed.

Reviewer #1 (Remarks to the Author):

The Authors submitted a remarkably extensive and detailed revised version which addresses all my comments in a satisfactory way. Although in general I agree with the Authors about the comments about the value of opportunistic screening, I am still convinced that this work has a significant but relatively limited clinical relevance and therefore I think that it would better fit a specialty journal about orthopaedics rather than Nature Communications. However, it is technically very sound, professionally presented and relevant to a wide audience and I have therefore no objections against its publication in this journal.

Answer: Thanks for your comments.

Reviewer #2 (Remarks to the Author):

The revised paper has addressed almost all my concerns. Several issues have been clarified and a couple new experiments have been done to further validate the method. A more thorough literature review has been given. A new experiment was done on GE DXA data. A performance comparison was done on different radiograph machines.

Answer: Thanks for your comments.

Reviewer #3 (Remarks to the Author):

The authors have addressed my comments appropriately.

I have also checked on the code/dataset but found some in-accuracy in the outcome regarding landmarks for both spine and hip.

I will mention them to the authors underneath as it can be address or at least commented in the papers.

Frankly speaking, the work is nice and well done but I do not know what is your "acceptance threshold" for your journal.

I would be tempted to accept as it is with the minor comments addressed.

Answer: Thanks for your comments. After reviewing the images again, we found 3 excluded hips containing both fractures and implants but only implants were reported. For the spine, the lower margin of one excluded vertebra covers part of the next vertebrae. In both conditions, the exclusion did not affect the BMD inference.

Some of the VCFs are borderline according to the Genant criteria. Our models only detected moderate to severe compression fractures to avoid ambiguity in determining mild or borderline deformities. The discussion on the VCF detection is detailed in the methods under the subtitle of 'Quality assessment of spine radiographs.'